# Study of Passive Adjustment Performance of Tubular Space in Subway Station Building Complexes

**Junjie Li [1], Shuai Lu [2,*], Qingguo Wang [3], Shuo Tian [1] and Yichun Jin [1]**

[1]  School of Architecture and Design, Beijing Jiaotong University, Beijing 100044, China;
    lijunjie@bjtu.edu.cn (J.L.); 17121721@bjtu.edu.cn (S.T.); 18121731@bjtu.edu.cn (Y.J.)
[2]  School of Architecture and Urban Planning, Shenzhen University, Shenzhen 518060, China
[3]  China Design and Research Group, Beijing 100042, China; wangqg@cadg.cn
*   Correspondence: Lyushuai@szu.edu.cn; Tel.: +86-10-18500234716; Fax: +86-10-62785691

**Abstract:** The stereo integration of subway transportation with urban functions has promoted the transformation of urban space via extensive two-dimensional plans to intensive three-dimensional development. As sustainable development aspect, it has posed new challenges for the design of architectural space to be better environmental quality and low energy consumption. Therefore, subway station building complexes with high-performance designs should be a primary focus. Tubular space is a very common spatial form in subway station building complexes; it is an important space carrier for transmitting airflow and natural light. As such, it embodies the advantages of effectively utilizing natural resources, improving the indoor thermal and light environments, refining the air quality, and reducing energy consumption. This research took tubular space, which has a passive regulation function in subway station building complexes as its research object. It firstly established a scientific and logical method for verifying the value of tubular space by searching causal relationships among the parameterized building space information factors, occupancy satisfaction elements, physical environment comfort aspects, and climate conditions. Secondly, based on the actual field investigation, a database of physical environment performance data and users' subjective satisfaction information was collected. Through the fieldwork results and analysis, the research thirdly concluded that the potential passive utilization of tubular space in subway station building complexes can be divided into two aspects: improvement in comfort level itself and utilization of climate between natural or artificial. Finally, three typical integrated design method for tubular spaces exhibiting high levels of performance and low amounts of energy consumption in subway station building complexes was put forward. This interdisciplinary research provides a design basis for subway station building complexes seeking to achieve high levels of performance and low amounts of energy consumption.

**Keywords:** passive space design; tubular space; physical building environment; fieldwork test; subway station building complex

---

**Highlights:**

(1) Construction of a multi-criteria analysis framework to analyze the passive adjustment performance of tubular space in subway station building complexes;

(2) Establishment of a database of physical environment performance and occupants' subjective satisfaction, based on actual field investigations;

(3) Development of an integrated design idea for tubular spaces in subway station building complexes that displays a high level of performance and low amount of energy consumption as the target orientation; and

(4) Proposal of three typical design concepts for compound tubular space.

# 1. Introduction

## 1.1. Research Background

With the rapid evolution of urban construction, the Transit-Oriented Development (TOD) mode has gradually formed a new organizational model for urban public spaces [1–3]. With the expansion of city subway, subway station building complexes have also entered a period of reinvention [4,5]. The stereo integration of subway transportation with urban functions has promoted the transformation of urban spaces from extensive two-dimensional plans, to intensive three-dimensional development [6]. Due to China's rapidly-advancing urbanization, the demand for sustainable development is becoming more and more urgent [7,8], and the issues of improving occupant comfort and reducing environmental load must be optimized [9,10]. The significant flow rate of people mainly in pass-through mode has led to lower environmental quality in above- and underground spaces at the junctions in subway station, and this may directly affect occupant comfort [11] and health [12]. In addition, large and complex public buildings tend to occupy a significant proportion of a city's energy consumption, threatening the sustainable development of human living environments [13,14].

## 1.2. Passive Design of Tubular Space in Subway Station Building Complexes

Passive design, which affects the sustainability of architecture from the prototype stage onward, is an important aspect of green building design [15]. Passive building design does not rely on active system equipment, but it does depend on a strong capacity for climate adaptability and self-adjustment, which creates a harmonious indoor coexistence of people and the outdoor environment [16,17]. Passive architecture describes buildings that are designed to cope with climate factors by providing enduring and natural comfortable indoor conditions [18,19]. The term "passive" conveys the idea of self-defense or self-protection of users in architectural design, with respect to the local natural environment [20,21]. A quality passive design avoids the possibility of high levels of energy consumption, saving up to 50% over traditional methods [22]. Therefore, the architectural prototype generally determines the degree of sustainability of the building.

Tubular space includes horizontal and vertical corridors in buildings, usually in slender shape, such as ventilation shafts, patios and lighting tubes, and tunnel corridors for connection. Tubular space occupies an important proportion of buildings in subway station building complexes, and its passive regulation has not been deeply investigated [23]. Subway station building complexes are affected by the characteristics of the mode of space utilization, wherein it is very common to use tubular space in ground-level and underground spaces (as shown in Figure 1), including patios and lighting tubes to improve natural lighting efficiency. Other uses include ventilation shafts for improving the indoor thermal environment and air quality, a station's traffic tubes for connecting ground-level and underground stations, and tunnels for traffic transmission. Tubular space can be seen as a "communications device" that transmits people, mass, and energy to different spaces [24]. This space type is a passive adjustment strategy located between the external and interior environments of the building. It uses natural energy sources (such as wind, solar energy, and rainwater) and the natural environment to regulate microclimates and improve the indoor atmosphere. In subway station building complexes, tubular space has the potential to play an important role in passive adjustment performance, especially with regards to natural lighting and ventilation, passive cooling, etc., to optimize comfort and user satisfaction with the indoor space, and greatly reduce the energy consumption of the building's operating phase [25].

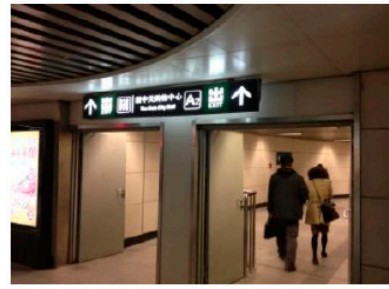 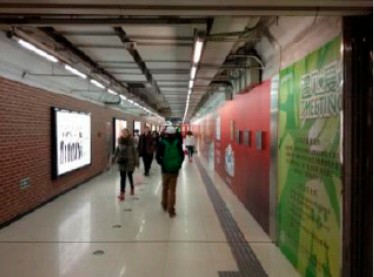 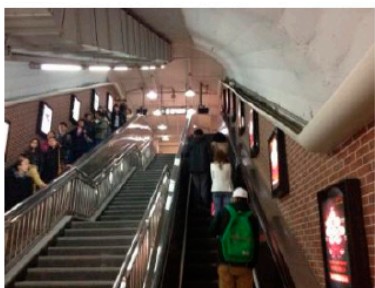

**Figure 1.** Examples of tubular spaces in subway station building complexes.

In their preliminary analyses of spatial design and climatic contradiction factors, some scholars have considered the particularity of using underground spaces over ground-level spaces, with respect to climate [26,27]. From the perspective of the physical environment of underground spaces, the degree of thermal and light comfort are of great importance. The comfort level of subway transit spaces and vehicle interiors has been verified by the coupling of actual tests with digital simulations [28]. After six years of actual tests of underground civil air defense space, Yong Li argued for a suitable acceptable thermal temperature range for underground areas [29]. In recent years, scholars have paid more attention to occupant health and placed a greater emphasis on ventilation and air quality, by conducting typological studies, and control and defense research on pollution and particulate matter (PM). Min Jeong Kim and others have proposed ventilation systems that can improve the platform PM10 levels and reduce ventilation energy, as compared to manual systems [30]. Practices based on this theory can be found as early as in ancient Rome, where ancient architects used tubular space to create more comfortable living environments. For instance, the underground corridor is a very good example of a kind of air cooling system in use at this time [31]. In terms of modern urban architecture, Hikarie Shibuya, a Japanese transportation complex designed as an integration concept that considers the passive utilization relationship between subway station buildings and above-ground structures, solved the problem of subway station lighting and ventilation [32]. In the retrofitting of the Les Halles area of Paris, the utilization of tubular space was adopted in underground spaces to provide natural lighting [33].

*1.3. Objective of this Study*

This study addressed a variety of forms of tubular space in city subway station building complexes, screening those spaces for passive adjustment potential in order to study the effect on the comfort level of the indoor environment and overall energy consumption. Through a fieldwork evaluation of building performance and occupant satisfaction in actual built projects, this research conducted a quantitative performance evaluation of passive architectural design strategies for tubular spaces. Through an analysis of the current situation and excavation of the spatial potential, this work determined the passive adjustment performance effects for subway station building complexes in terms of sustainable development, therefore providing a basis for improving architectural design methods to show higher levels of performance and lower amounts of energy consumption. This research pursued the following three objectives: (1) provide a basis for design by analyzing the types of passive function and specific variables for the technical strategies employed by subway station building complexes; (2) test the passive adjustment effects of subway station building complexes and improve the authenticity and objectivity of the designs via actual and effective environmental monitoring evaluation; and (3) explore typical further-optimized strategy models for the compound tubular space systems of subway station building complexes, and provide guidance for design optimization.

**2. Methodology**

This research was based on the dual perspectives of architecture and the built environment. With regards to architectural design, this work produced a space prototype and deconstructed the

factors affecting the passive adjustment performance of architecture. According to a comparison of the factors that influence the quality of indoor buildings and actual built environments, a comprehensive evaluation was made of the passive adjustment performance of tubular spaces in subway station building complexes [34]. First, the factors that affect the passive adjustment performance were analyzed. Then, according to the analytic factors and taking the urban Beijing subway station building complexes as an example, a long-term physical environment test was carried out. The subject was a subway station building complex with a typical amount of tubular space. This research focused on the physical environment, as tube as the passive function of potential space and its influence on surrounding functional areas. It included actual measurement results such as the thermal conditions, air ventilation, lighting environments, indoor air quality, occupant satisfaction and comfort, and other subjective feedback. Through this objective investigation of the physical environment and subjective feedback of the occupants' degree of comfort, the problems with objective space were able to be studied and analyzed, and the potential for spatial optimization put forth from the perspective of passive adjustment performance. Finally, the database established through this research assisted in highlighting the design goals for tubular space in subway station building complexes. A model for three typical kinds of composite tubular spaces was constructed with the goal of achieving high levels of performance and low amounts of energy consumption. Therefore, this research method was divided into the following four steps. (as shown in Figure 2)

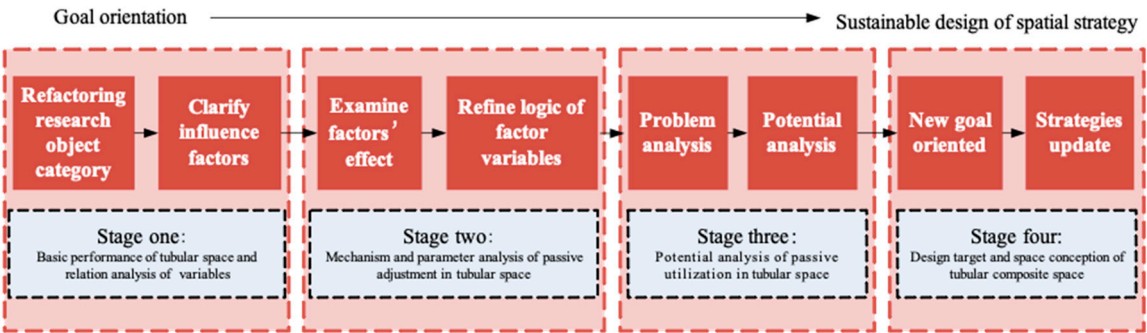

**Figure 2.** Research methodology route.

*2.1. Stage One: Factor Analysis of the Effect of Passive Design on Tubular Space*

Passive design belongs to the category of sustainable development in architecture, and is an essential part of addressing three factors: the environment, society, and the economy [35]. Tubular space is a typical passive spatial design strategy that coordinates contradictions among these three factors and architecture, leading building construction in a more positive direction. Architecture is a carrier of the climate and its human occupants; building space and outdoor climate conditions can be seen as the reasons for indoor physical environments and occupant satisfaction [36]. This research is based on an AHP (Analytic Hierarchy Process) methodology which decomposes complex issues into several group factors, and compares those factors to one another to determine their relative importance [37,38]. It adopted the method of factor quantification analysis to classify climate conditions, building spaces, physical environmental comfort, and occupancy satisfaction. This logical framework was established through measurements and simulations; the influences therein were determined by a correlation analysis, based on the acquired data. Therefore, the passive design of tubular space was divided into four factor groups: spatial parameters, climate parameters, the degree of physical environmental comfort, and occupancy satisfaction. (as shown in Figure 3)

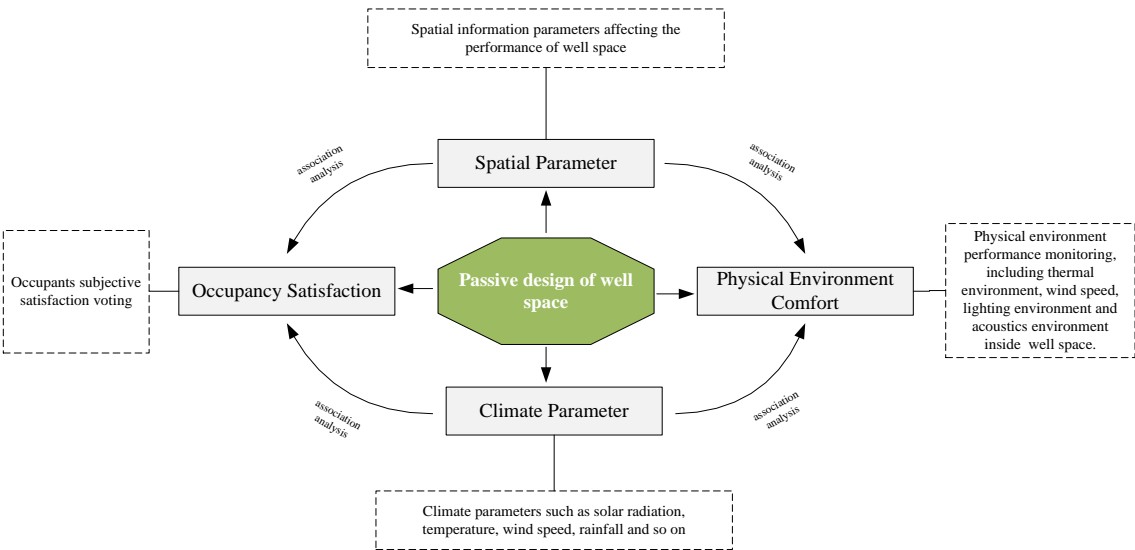

**Figure 3.** Analysis of the factors influential on tubular space in subway station building complexes.

According to authors pervious research in passive space design [36,39], the building space can be recognized as four categories including "shape", "mass", "quantity", and "connection". This research also adopts this research framework that the factor groups were further quantified and subdivided. The spatial parameters were again divided into four sub-factors: geometric dimensions, interface properties, and internal and external related categories (as shown in Table 1). Researchers have made useful contributions to indoor environment comfort research [40]. For example, Professor Fanger highlighted six elements that influence the comfort level of an indoor thermal environment (the mean radiation temperature, air temperature, relative humidity, air speed, clothing insulation, and metabolic rate) to form the Predicted Mean Vote (PMV) model [41]. LEED [42] in USGBC and China ESGB [43] lists lighting environment evaluation factors (including natural light, artificial light, view, and control) and offers clear standard indicators. As the tubular space can have a significant wind effect that affects the indoor air flow, while most tubular space exists underground, and the outdoor noise environment has little influence on the interior, therefore, outdoor climate parameters can be divided into four additional sub-factors: thermal environment, lighting environment, air quality, and wind speed. Corresponding to the outdoor climatic conditions, the indoor physical environment also includes four sub-factors: thermal environment, lighting environment, air quality, and air velocity. There are many factors involved in occupancy satisfaction. The method of semantic differential (SD) analysis [44] was used to divide the influencing factors into eight aspects: thermal conditions, humidity, light, air quality, air velocity, ease of use, cleanliness and maintenance, and overall environmental satisfaction (as shown in Table 1).

**Table 1.** Factor analysis of the passive adjustment effect of tubular space.

| Factor Group | Factors | Parameter Acquisition Method | Parameter Unit |
|---|---|---|---|
| Spatial parameter | geometric dimensions (L:W:H) | distance measurement | m |
| | interface property (U-value) | material thermal performance calculation | $W/(m^2 \cdot K)$ |
| | internal related categories (visitor flow rate) | statistics | N/h |
| | external related categories (outdoor, platform, commercial, none) | judgment | N/a |
| Climate parameter | thermal environment (temperature, relative humidity) | measurement | °C, % |
| | lighting (illuminance) | | Lux |
| | air quality (PM2.5, PM10, $CO_2$) | | $\mu g/m^3$, ppm |
| | wind speed | | m/s |
| Degree of comfort with the physical environment | thermal environment (temperature, relative humidity) | | °C, % |
| | lighting (illuminance) | | Lux |
| | air quality (PM2.5, PM10, HCHO, $CO_2$) | | $\mu g/m^3$, ppm |
| | air velocity, wind temperature | | m/s, °C |
| Occupancy satisfaction | thermal comfort | occupant survey | Vote score [−3~3] |
| | humidity | | |
| | air quality | | |
| | lighting | | |
| | ventilation | | |
| | ease of use | | |
| | cleanliness and maintenance | | |
| | overall environmental quality satisfaction | | |

## 2.2. Stage Two: Field Survey

Corresponding with the public factors affecting the passive function of tubular space, the actual field investigation involved spatial drawings, monitoring the indoor and outdoor physical environments, and determining occupants' subjective levels of satisfaction. The relationship between buildings and people, especially in terms of the healthiness of the indoor environment, has a significant influence on human survival and sustainable development. Since the normal operating phase tends to be from 6:00 a.m. to 10:00 p.m., the opening time of building complexes (offices or businesses) is usually included within that time frame. Therefore, the survey had a clear research plan regarding a day cycle time, from 6:00 a.m. to 10:00 p.m., including two rush hours where there was peak human flow. The physical environment test called for the selection of a typical space, such as tube entrance, middle tubular space, connection point between a subway and complex building, station hall, or subway platform where the long-term physical environment could be monitored. The data were collected every five minutes. The physical quantities included nine parameters: temperature, humidity, illuminance, $CO_2$ concentration, PM2.5, PM10, HCHO, air velocity, and wind temperature. The test contents including long-term consecutive days of outdoor temperatures which measurement interval was 5 min, temperature measurements for each (selected) test point which measurement from 6:00 a.m. to 10:00 p.m. for each typical day; measurement interval was 5 min.

In addition, since piston wind can affect the interior tubular space and connection points of urban complexes during subway operation, and the aerodynamic forces of piston wind may be usable as a source of renewable energy [45], the fieldwork test also included instantaneous wind speed changes at

the subway platform level. The observation frequency was 3 s, with a train cycle of arrival, stay, and departure (as shown in Table 2).

**Table 2.** Building the physical environment fieldwork test framework.

| Measurement Items | | Parameter Type | Test Content | Properties of the instruments |
|---|---|---|---|---|
| Thermal environment | outdoor temperature test | temperature | °C | Portable infrared temperature meter, Biaozhi GM700, Range: −50~700 °C, Resolution: 0.1 °C |
| | indoor temperature test for each (selected) test point | | | |
| | indoor humidity test for each (selected) test point | humidity | % | |
| Lighting | outdoor luminance test | luminance | lux/daylight factor % | Portable luminance meter, Reggiani DT-1301, Range: 0~50,000 Lux, Resolution: 1 Lux |
| | indoor luminance test for each (selected) test point | | | |
| IAQ | outdoor $CO_3$ concentration test | $CO_2$ concentration | ppm | Portable and self-record $CO_2$ meter, TJHY-EZY-1, Range: 0~5000 ppm, Resolution: 1 ppm |
| | indoor $CO_2$ concentration test for each (selected) test point | | | |
| | outdoor PM2.5/10 concentration test | PM2.5/10 concentration | $\mu g/m^3$ | Portable air quality meter, temopt LKC-1000S+, Range: 0~999 $mg/m^3$, Resolution: 0.01 $mg/m^3$ |
| | indoor PM2.5/10 concentration test for each (selected) test point | | | |
| | indoor HCHO concentration test for each (selected) test point | HCHO | $g/cm^3$ | |
| Ventilation | indoor air velocity test for each (selected) test point | air velocity | m/s | Self-record instrument for wind velocity and wind temperature, TJHY-FB-1, Range: 0~10 M/S, Resolution: 0.01 M/S |
| | indoor air temperature of each (selected) test point | air temperature | °C | Self-record instrument for environment, TJHY-HCZY-1, Range: 0~5000 ppm, Resolution: 1 ppm |

The object of this investigation was the tubular spaces in a subway station building complexes, so the function and shape of the spaces were simpler and more explicit than other common building spaces. The design of the subjective questionnaire focused on the degree of occupancy comfort and the space's influence on human health during short stays. In general, the questionnaire included three categories: satisfaction with the physical environment; space satisfaction votes, such as ease of use, cleanliness, and maintenance; and overall satisfaction with the environment's quality.

In order to improve the efficiency of the investigation, spatial drawings, subjective satisfaction research, and studies of comfort related to the objective physical environment were all conducted. The characteristics of the subway space in utilization mode were special in that mostly the area was designed for a rapidly-passing crowd who would be present only for a short stay. The connection spaces often fluctuated in terms of the physical environment, and users' moods and physical health conditions likely varied considerably. Therefore, the method of subjective investigation also needed to be more diverse than in other similar types of research. The questionnaire was collected mainly by three means: website-based and on-site surveys, and on-site interviews. The purpose of the website-based questionnaire was to avoid misunderstandings related to temporally-subjective factors, and dispel individual elements through the long-term accumulation of memory. For the satisfaction and self-reported productivity questions, the survey used a 7-point semantic differential scale with endpoints of "very dissatisfied" and "very satisfied." For the purposes of comparison, the scale was assumed to be roughly linear, with ordinal values for each of the points that ranged from −3 (very dissatisfied) to +3 (very satisfied) and 0 as the neutral midpoint [46]. The on-site interviews were with specifically-selected occupants who stayed in the space for a long period of time, such as retail vendors, subway station operators, commercial building security, cleaning and maintenance staff, etc. The research methods were not rigidly obeyed for prescribed problems and formats, and

were altered via conversations to help the researchers understand the respondent's age, cultural and economic background, space satisfaction, environmental problems, etc (as shown in Table 3).

**Table 3.** Occupancy satisfaction voting framework.

| Test Items | | Parameter Type | Test Content | Test Content |
|---|---|---|---|---|
| Physical environment satisfaction | thermal comfort | vote | web-based survey/ fieldwork-based survey/ human perception test | 7-point scale [−3,−2,−1,0,1,2,3] very dissatisfied to very satisfied |
| | humidity | | | |
| | air quality | | | |
| | lighting | | | |
| | ventilation | | | |
| Space satisfaction | ease of use | vote | web-based survey/ fieldwork-based survey/ human perception test | 7-point scale [−3,−2,−1,0,1,2,3] SD of feelings about space's atmosphere |
| | cleanliness and maintenance | | | |
| Overall environmental quality satisfaction | | vote | web-based survey/ fieldwork-based survey/ human perception test | 7-point scale [−3,−2,−1,0,1,2,3] very dissatisfied to very satisfied |

### 2.3. Stage Three: Problem and Analysis of the Spatial Potential

Stage two set up a framework for a comprehensive system that focused on three factors: architecture, humans, and the environment. Based on the conclusions made during that stage, the passive strategy factors affecting the research object were classified and recombined to analyze the functional characteristics of different positions of space that can be found throughout a subway station building complexes. According to the research by Margarita N. Assimakopoulos regarding the thermal environment in Greek subways [47], Teresa Moreno and colleagues' work on airborne particulate matter in the Barcelona subways, and John Burnett [48] and associates' investigation and analysis of the lighting environment in the Hongkong metro in China [49], obvious problems such as high humidity, low thermal comfort, and poor air quality and lighting environments in subway halls and platform spaces all emerged as worthy of further research. Many scholars also argued for energy-saving strategies in subway systems by means of passive ventilation designs for complete ventilation systems, for instance by developing ventilation systems in subway stations that could control indoor air pollutants [50,51]. The core of the third stage of this study includes two aspects. First, through an investigation and analysis of the status quo, the existing environmental problems were extracted and a design strategy put forward to resolve certain issues. Secondly, through statistical data, the researchers discovered the potential capacity of passive space, and identified design opportunities that could improve comfort, health, and energy efficiency.

### 2.4. Stage Four: Set up New Target Orientation and Space Update

The fourth step of this research put forward the spatial design goal of tubular space in subway station building complexes, from the perspective of sustainable development. Certain space assumptions and a particular design procedure for the compound tubular space were promoted to provide guidance for the optimized design.

Through data and problem analyses and potential excavation, this research searched typical models of complex tubular space complex systems that would be applicable to subway station building complexes. The key point was to determine an applicable and feasible space utilization model that could provide a design basis. The researchers put special emphasis on viable applications for potential natural resources in subway spaces, such as tunnel and piston wind, pull shafts, lighting, and landscape tubes, that could be further integrated into the design. Typical composite tubular spaces can be in the form of a tube tunnel (a solar chimney composite space system), combined wind tunnel (a displacement

ventilation complex space system), combined active and passive ground source heat pump (wind tunnel composite tubular space system), hot air shaft ventilation, or lighting composite space system. Figure 3 offers an overall view of the study.

## 3. Results and Discussion

### 3.1. Building Space Information Factors

Based on the above-mentioned factors, a multi-criteria evaluation method for tubular space was proposed. The survey selected five typical subway station building complexes in Beijing, including Xizhimen (W1), Haidianhuangzhuang (W2), Guomao (W3), Dawanglu (W4), and Wangfujing Stations (W5) (as shown in Figure 4). The selected five stations involved six main subway lines: #1, #2, #4, #10, #13, and #14. All of the surveyed stations were urban subway transit hubs. Xizhimen Station (W1) is where the #2, #4, and #13 subway lines converge; it is also home to three high-rise commercial office buildings and the Beijing subway station. Haidianhuangzhuang (W2), Guomao (W3), and Dawanglu Stations (W4) all are points of convergence for more than two subway lines, and offer connections with urban complexes. Wangfujing Station (W5) is located in the middle of Beijing, and is connected to the largest integrated commercial building in Asia. As such, it features a substantial people flow rate; the location is also of geographical importance (as shown in Table 4).

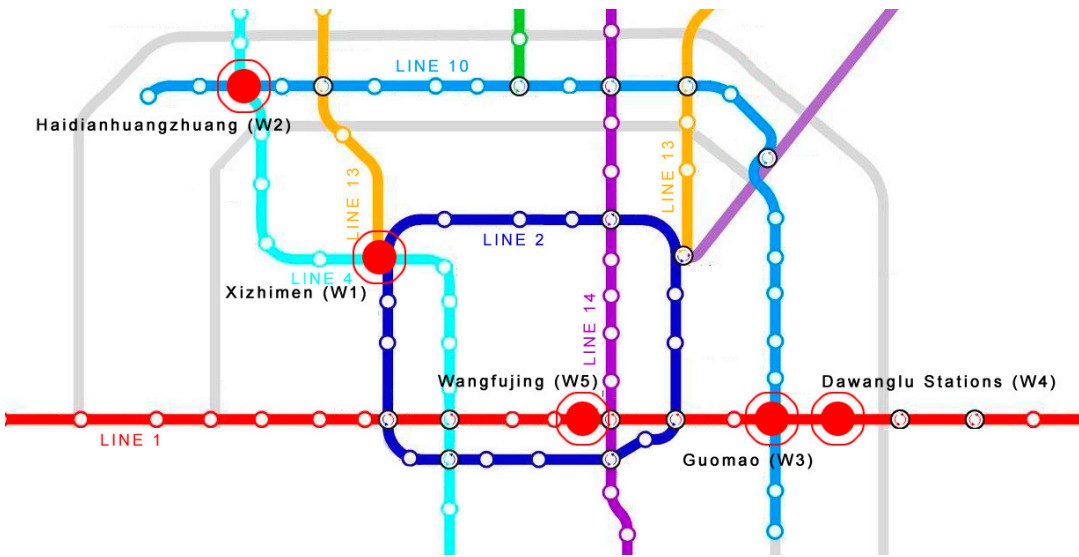

**Figure 4.** Five typical subway station building complexes in Beijing.

The test period was selected to be from 28th June 2017 to 20th July 2017, the highest-temperature time period in Beijing. It is a typical and continuous testing period of 3 weeks. The data excluded unstable factors which may conduct to instantaneous mutations data such as weather mutations, active equipment interference, people behavioral interference, misuse of testing instruments, etc, and used a mean value within the 3 weeks. The purpose of this research was to investigate the performance of the tubular space in the physical environment under the most unfavorable conditions in the summer climate.

**Table 4.** Survey object information.

| No. | Station Name | City Complex | Building Function | Building Area (m²) | Subway Line | Number of Test Points | Type of Test Point |
|-----|--------------|--------------|-------------------|--------------------|-------------|-----------------------|--------------------|
| W1 | Xizhimen | Cade mall | Commercial, office | 89,000 | #2, #4, #13 | 3 | middle tubular space, tube entrance, platform layer |
| W2 | Haidianhuangzhuang | Gate City mall | Commercial | 47,000 | #4, #10 | 3 | middle tubular space, tube entrance |
| W3 | Guomao | Yintai mall | Commercial, office | 350,000 | #1, #10 | 4 | middle tubular space, tube entrance, platform layer, station hall |
| W4 | Dawanglu | China Trade Center mall | Commercial, office | 710,000 | #1, #14 | 4 | middle tubular space, tube entrance, platform layer, station hall |
| W5 | Wangfujing | Oriental Plaza mall | Commercial | 120,000 | #1 | 4 | middle tubular space, tube entrance, platform layer, station hall |

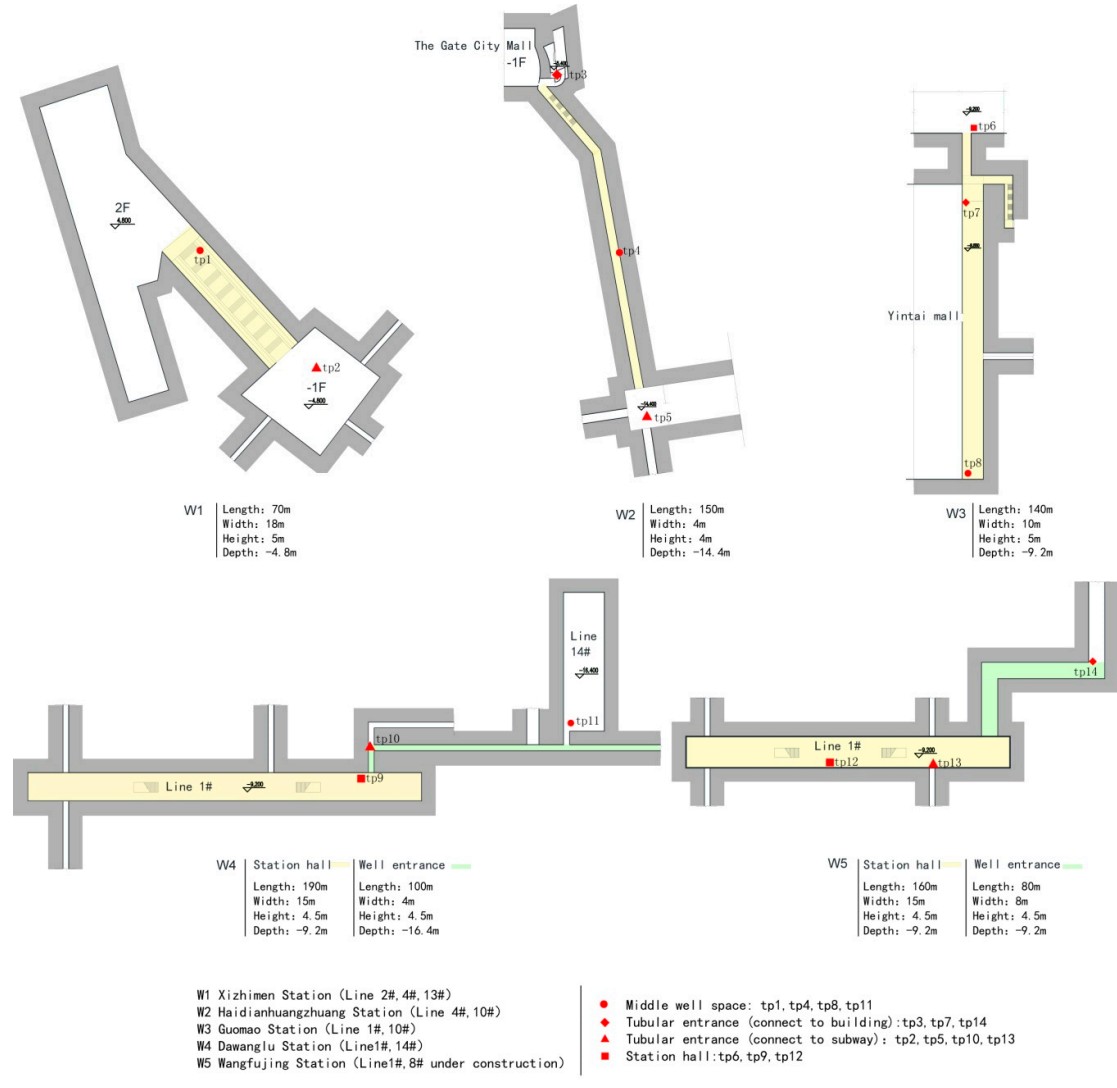

**Figure 5.** Test station space plan and test points location.

The first step in the fieldwork survey was to acquire the building information. This was performed in order to obtain the data supporting the tubular spaces. After the fieldwork test, the study first

drew out five site plans and geometric scale parameters for the tubular space (as shown in Figure 5). The researchers then selected three or four test points for each site station; all tests contained 18 measuring points. Each test point had a certain representativeness. The W1 site contained three test points; tp1 was located in the middle of an above-ground glass corridor between the station and the complex, and was a middle tubular space.Tp4 and tp11 were located between the sites and underground associated tubes, which were all classified as middle tubular space. Tp8 was a link between a subway station and commercial building, which was also middle tubular space. Tp3, tp7, and tp14 belonged to the first kind of tube entrance space, and connected the commercial complex building at one end of the tube shaft. Tp2, tp5, tp10, and tp13 belonged to the second kind of tube entrance space, and connected to the station hall. Tp6, tp9, and tp12 were the test points at the subway station hall (as shown in Figure 5). Finally, tp15, tp16, tp17, and tp18 were the subway platform test points for W1, W3, W4, and W5, respectively.

Because the locations of the tubular spaces in each subway station building complexes were different, the environmental problems varied dramatically. Therefore, this research compared the physical environment parameters of the 18 measuring points, according to five types: middle tubular space (four test points), tube entrance (connection to building) (three test points), tube entrance (connection to subway) (four test points), station hall (four test points), and platform layer (four test points).

### 3.2. Field Survey Results and Analysis

### 3.2.1. Physical test results and analysis

Professor Fanger highlighted six elements that influence the comfort level of an indoor thermal environment (mean radiation temperature, air temperature, relative humidity, air speed, clothing insulation, and metabolic rate); all six are necessary to form the Predicted Mean Vote (PMV) model [41]. Tubular space in subway station building complexes is mostly underground, so the influence of radiation temperature can be ignored and people's metabolic rates can be set to the same level of 1.5 met [41]. During the test period, the hottest period in summer in Beijing was selected, so clothing level was chosen as 0.35 clo [41]. Data from this survey were collected according to eight physical parameters: environment temperature, humidity, illuminance, air velocity, PM2.5, PM10, and the HCHO and $CO_2$ concentrations at each point. The average result values are shown in Tables 5–9. According to the current national standards and norms, the typical range of thermal comfort is defined as between 16~28 °C [43], humidity comfort is between 30~60% [42], illumination should be no higher than 150 lux [52], and indoor air velocity in winter should be lower than 0.15 m/s and 0.25 m/s in summer. The concentrations of PM2.5 and PM10 should be lower than 75 μg/m$^3$ and 150 μg/m$^3$, respectively, according to the 24-hour average concentration limits of the two grades listed in the national standard [53]. The concentration of HCHO and $CO_2$ should be lower than 0.08 mg/m$^3$ [54] and 1000 ppm [55], respectively.

(1) Comfort analysis

Figures 6–9 are box diagrams of the physical test results for thermal conditions, lighting, IAQ, and ventilation in all five types of tubular space. The red zones show locations where the occupant comfort values were beyond the related comfort standard.

As regards the thermal environment, tp1 was a solar corridor space with a higher temperature than the other three points, which were at the boundary of the comfort zone (the point at which the human body would no longer enjoy thermal comfort). The tube lengths of tp4 and tp8 were 150 m and 140 m, respectively, and the humidity levels of the two test points significantly exceeded the standard. The highest reached 84.7%. The temperature in the space was lower than the human comfort level standard would deem acceptable, and the excessive humidity could easily cause mildew and affect users' health. As regards the lighting environment, the illuminance levels of tp4 and tp8 in the middle tubular space were not sufficient; the values did not reach the national standard requirement and

thus could be hiding potential dangers. Due to the large amount of natural light at tp1, the average illuminance could reach 990 lux without artificial lighting. A better lighting environment would also improve the quality of the indoor environment. Past research results have shown that occupants become uncomfortable and can even experience headaches and chest tightness, causing their work ability to decline, when the $CO_2$ concentration is over 1000 ppm. In the test of $CO_2$ concentration for the middle tube, the two longer tube sat tp4 and tp8 had lower air quality; the highest $CO_2$ concentration was at tp4, which reached 1931 ppm. This is over two times the standard. The maximum PM2.5 concentration at tp8 reached 121 g/m$^3$; the outdoor concentration was 74.6 g/m$^3$. Thus, the disadvantages were greatly exacerbated. Although the air velocity values at tp4 and tp11 were slightly higher than the standard, moderate ventilation in a thermal environment can lower body surface temperature, taking away the sweat that collects on the human body's surface.

To sum up, the main problems as regards comfort in the middle tubular space were its high humidity, poor light environment, and low air quality.

**Table 5.** Average values for the physical test results in the middle tubular space.

| Site No. | Test Point Number | Temperature (°C) | Humidity (%) | Illuminance (Lux) | Air Velocity (m/s) | PM2.5 | PM10 | HCHO | CO$_2$ |
|---|---|---|---|---|---|---|---|---|---|
| W1 | Tp1 | 32.1 | 47.7 | 990 | 0.196 | 47.9 | 67.2 | 0.022 | 681 |
| W2 | Tp4 | 28.1 | 70.2 | 72.0 | 0.265 | 45.4 | 64.3 | 0.030 | 1163 |
| W3 | Tp8 | 25.5 | 76.8 | 19.0 | 0.0 | 66.1 | 92.0 | 0.057 | 1000 |
| W4 | Tp11 | 28.3 | 57.4 | 153.6 | 0.307 | 52.0 | 73.1 | 0.044 | 892.1 |

Note: The data in the grey background indicate that it is beyond comfortable zone.

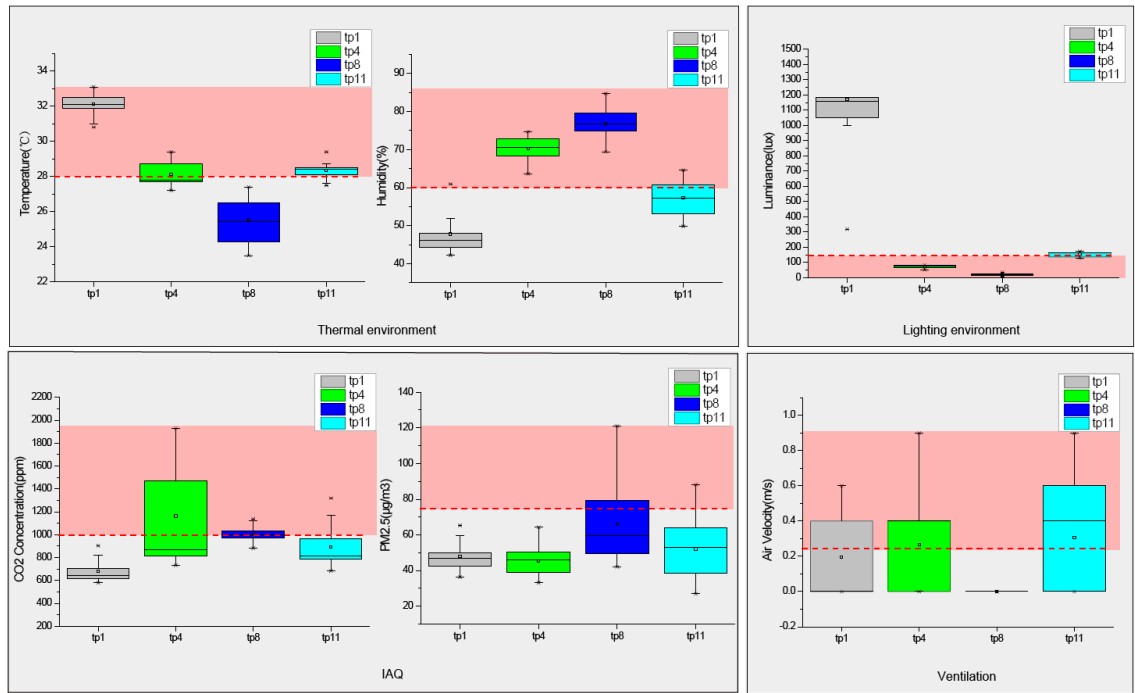

**Figure 6.** diagram of the physical test results from the middle tubular space.

Figure 7 shows the performance of the physical environment of the tube entrance space approach to one end of a shopping mall. All of the environment's temperatures and most of the humidity index values for the three test points exceeded the comfort zone. The maximum value of tp3 was 32.3 °C, while the mean outdoor temperature was 29 °C. The average illuminance levels of the three test points in the tube entrance (connection to building) did not reach the national standard; tp3 and tp7 were especially low. Both of the test points located at the indoor and outdoor junctions had light

levels that would require human eyes a significant amount of time to adjust to the darkness. This can cause sensations of insecurity and discomfort when entering from a strong outdoor light environment. The concentration of $CO_2$ in these spaces was also high, with maximum values for tp3 and tp7 reaching 1761 ppm and 1690 ppm, respectively.

To sum up, because most of the tube entrance space was connected to a shopping mall and was close to the outdoors, the thermal environment and humidity levels were low, light problems were obvious, and concentrations of $CO_2$ were high.

**Table 6.** Average values for the physical test results from the tube entrance (connection to a building).

| Site No. | Test Point Number | Temperature (°C) | Humidity (%) | Illuminance (Lux) | Air Velocity (m/s) | PM2.5 | PM10 | HCHO | CO$_2$ |
|---|---|---|---|---|---|---|---|---|---|
| W2 | Tp3 | 30.6 | 62.1 | 7.5 | 0.117 | 54.8 | 76.3 | 0.055 | 981.9 |
| W3 | Tp7 | 30.4 | 65.2 | 24.4 | 0.144 | 73.5 | 102.8 | 0.126 | 1308.2 |
| W5 | Tp14 | 28.9 | 52.9 | 147.9 | 0.604 | 32.2 | 45.0 | 0.047 | 928.9 |

Note: The data in the grey background indicate that it is beyond comfortable zone.

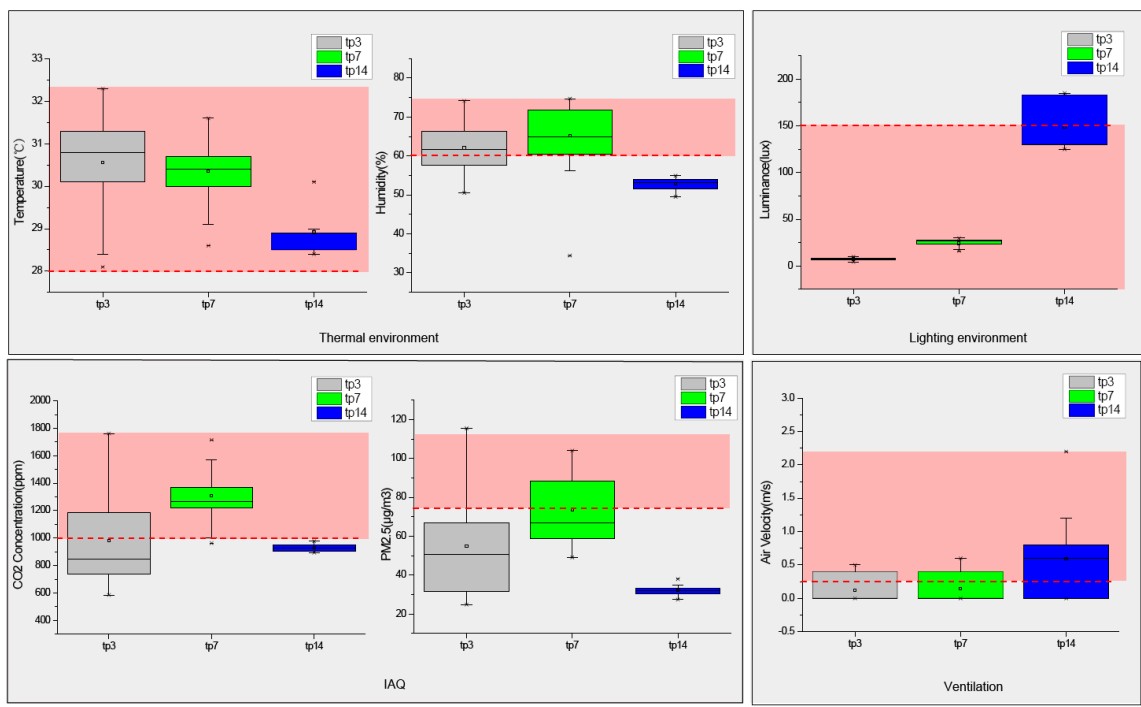

**Figure 7.** Box diagram of the physical test results from the tube entrance (connection to building).

According to Figure 8, the thermal and humidity environments at the tube entrance space near the subway were obvious. The temperatures at the four test points all exceeded the comfort level. The average temperature at tp5, with an elevation depth of 14.4 m, was 2.5 °C lower than that of tp10 and tp13, which were at an elevation depth of 9.2 m. The higher wind speed relieved the high humidity and $CO_2$ concentrations at the tube entrance space (connection to subway). Generally, the average humidity was lower than 70% and both the average and maximum values of $CO_2$ were greatly reduced. However, due to the influence of the subway piston wind, the wind speed presented a sinusoidal fluctuation and the direction of the wind speed changed periodically. Thus, the wind environment became the most unfavorable factor with regards to comfort.

To sum up, ventilation was one of the most detrimental comfort factors at the tube entrance space (connection to subway). It was affected by the piston wind so that cold air (from the air conditioning) was sometimes sucked out of the subway platform layer and hot air was pumped out of the tubular space. People who stayed at that location for long periods of time were frequently affected by two

kinds of wind that had large temperature and direction differences. They expressed great discomfort and likely were experiencing threats to their health.

**Table 7.** Average values for the physical test results from the tube entrance (connection to subway).

| Site No. | Test Point Number | Temperature (°C) | Humidity (%) | Illuminance (Lux) | Air Velocity (m/s) | PM2.5 | PM10 | HCHO | CO$_2$ |
|---|---|---|---|---|---|---|---|---|---|
| W1 | Tp2 | 30.8 | 50.0 | 154 | 0.365 | 53.5 | 75.9 | 0.019 | 688.4 |
| W2 | Tp5 | 28.6 | 66.0 | 32.5 | 0.859 | 52.7 | 73.6 | 0.019 | 840.5 |
| W4 | Tp10 | 31.3 | 68.3 | 107.3 | 1.73 | 107.6 | 151.1 | 0.021 | 920.8 |
| W5 | Tp13 | 31.1 | 70.0 | 91.2 | 0.683 | 149.2 | 201.7 | 0.026 | 802.4 |

Note: The data in the grey background indicate that it is beyond comfortable zone.

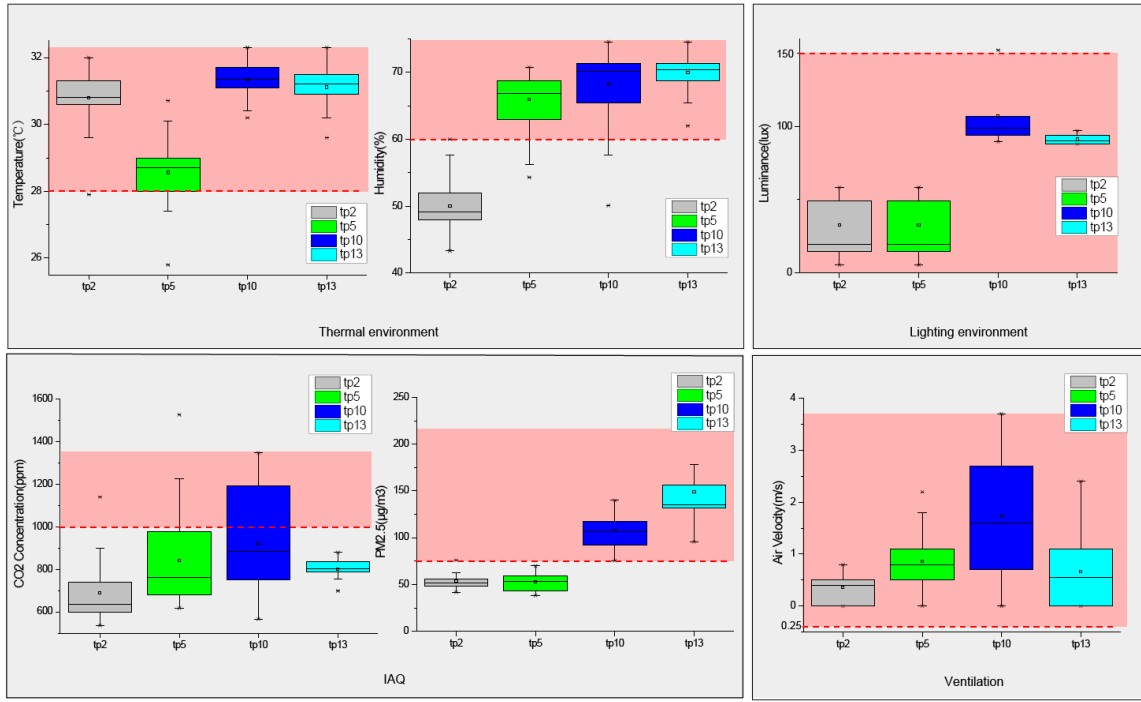

**Figure 8.** Box diagram of the physical test results at the tube entrance (connection to subway).

The physical environment test at the station hall focused on Beijing subway line #1, and included three test points. As can be seen from Figure 9, the environment temperature in the station hall was significantly higher than the comfort zone and 5 to 6 °C higher than the outdoor temperature at that time. The humidity in the station hall was also relatively high, but the light environment basically met the national standard. Due to a large number of people flowing through the middle of the station hall and the relatively longer tube lengths (130 m, 190 m, and 160 m), the CO$_2$ concentrations at the three points were correspondingly higher. The peak concentration of CO$_2$ at the tp6 measuring point reached 1833 ppm. Tp12 was located at a point with a smaller people flow rate, so its CO$_2$ concentration stayed within the standard range. However, due to the impact of the subway piston wind, there was a high level of PM pollution in the subway tunnel that was brought into the entrance hall, with a peak concentration of 160 g/m$^3$. Compared with the outdoor concentration 74.6 g/m$^3$, this was two times the outdoor concentration at that time. Thus, it can be seen that a large number of people were gathered at the station and hall levels, resulting in a higher concentration of CO$_2$. The station hall layer was directly connected to the train platform and had a higher PM concentration due to the influence of the piston wind from the subway.

In summary, influenced by the space size and flow rate, the physical environment of the hall at all stations was the worst, which is reflected in the high temperature and humidity, and poor air quality.

**Table 8.** Average values of the physical test results from the station hall.

| Site No. | Test Point No. | Temperature (°C) | Humidity (%) | Illuminance (Lux) | Air Velocity (m/s) | PM 2.5 | PM 10 | HCHO | CO$_2$ |
|---|---|---|---|---|---|---|---|---|---|
| W3 | Tp6 | 33.5 | 62.2 | 144.1 | 0.724 | 68.6 | 96.0 | 0.038 | 1374.4 |
| W4 | Tp9 | 32.6 | 61.9 | 152.3 | 0.446 | 100.9 | 139.2 | 0.023 | 1153.6 |
| W5 | Tp12 | 31.2 | 71.4 | 92.0 | 0.172 | 119.6 | 167.2 | 0.021 | 922.5 |

Note: The data in the grey background indicate that it is beyond comfortable zone.

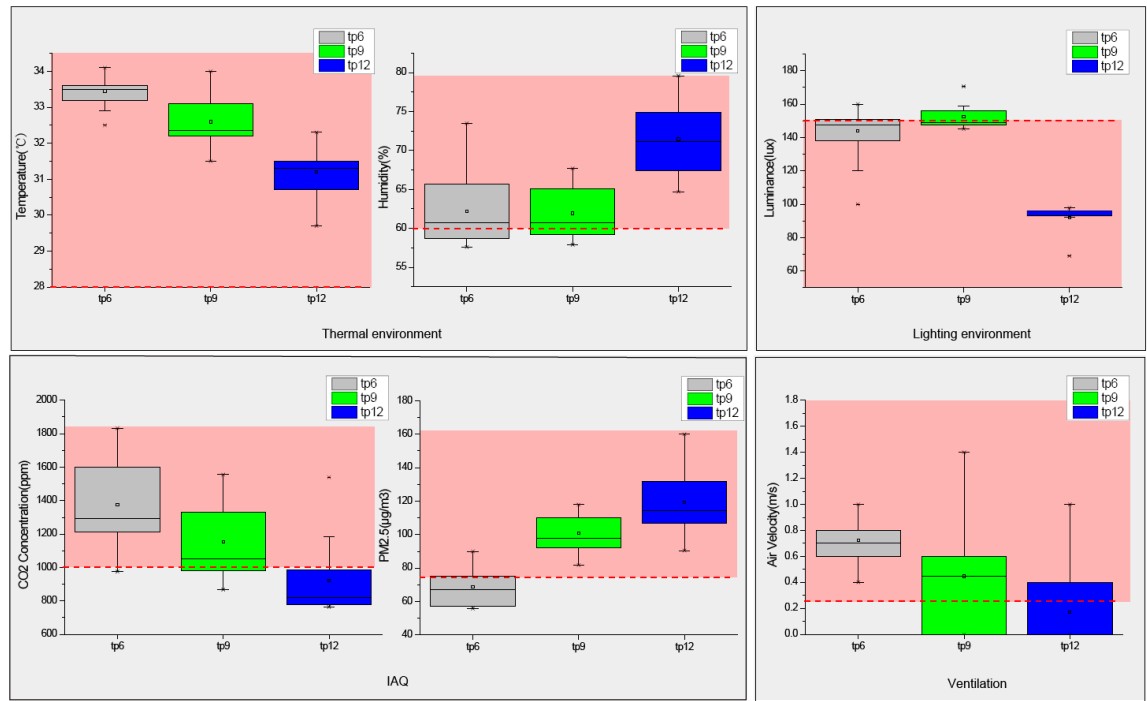

**Figure 9.** Box diagram of the physical test results from the station hall.

The physical environment of the platform layer is shown in Figure 10. Since the wind speed at the platform layer was significantly affected by the movement of the subway vehicles, it will be discussed in detail in the next chapter. The lighting environment met the national lighting standards. There were two key problems with comfort: the thermal environment and air quality. The temperature was generally too high; the highest value was from W4, which reached 34.5 °C. The values were 3.5 °C higher than the outdoor temperature at that time. For air quality, the most significant problem was the PM concentration. The PM2.5 data for almost all of the test sites were higher than the human body's comfort range; the PM10 concentration was too high at the W4 site as tube, reaching a maximum of 225.7 g/m$^3$, while the outdoor concentration was 83.2 g/m$^3$.

To sum up, the temperature and PM values were the key problems with comfort at the platform layer.

**Table 9.** Average values for the physical test results from the platform layer.

| Site No. | Test Point No. | Temperature (°C) | Humidity (%) | Illuminance (Lux) | Air Velocity (m/s) | PM2.5 | PM 10 | HCHO | CO$_2$ |
|---|---|---|---|---|---|---|---|---|---|
| W1 | tp15 | 30.2 | 48 | 260.3 | Instantaneous | 85.3 | 121.7 | 0.017 | 974.5 |
| W3 | tp16 | 31.6 | 64.3 | 263 | wind speed | 98.5 | 138.3 | 0.019 | 867 |
| W4 | tp17 | 33.9 | 58.8 | 259.5 | (as shown in | 137.6 | 192.5 | 0.02 | 1005 |
| W5 | tp18 | 33.7 | 69.6 | 251.2 | Figure 12) | 72.7 | 102.3 | 0.019 | 1283 |

Note: The data in the grey background indicate that it is beyond comfortable zone.

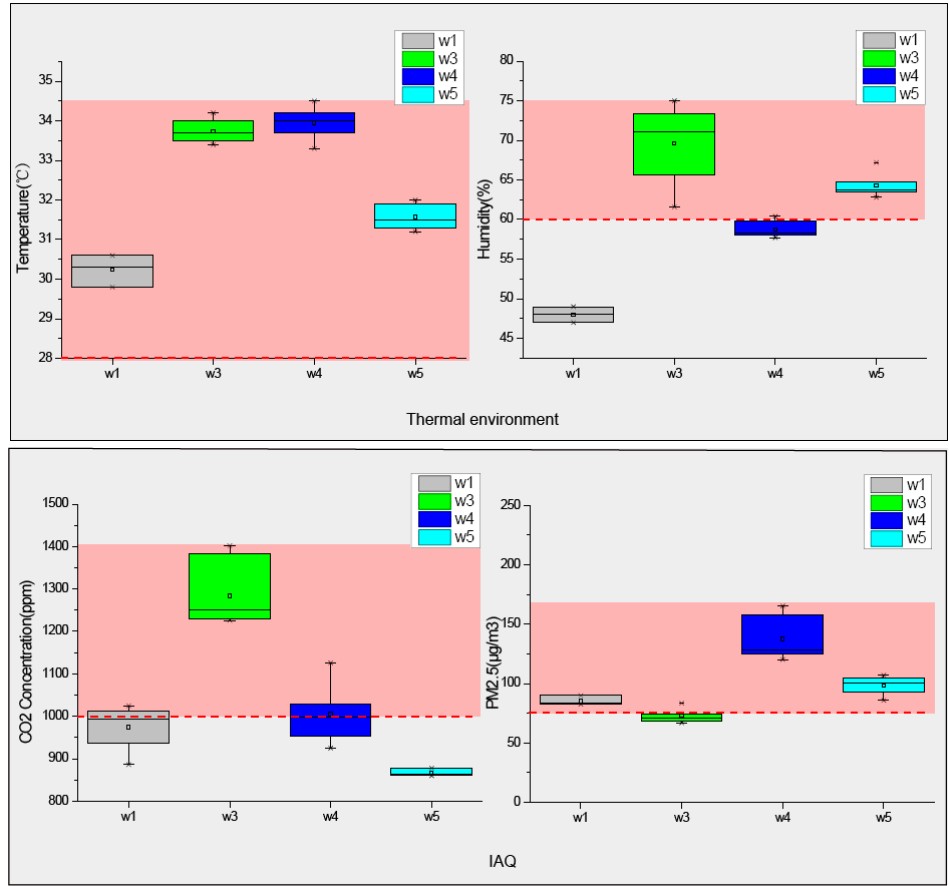

**Figure 10.** Box diagram of the physical test results from the platform layer.

(2) Changes in physical quantities with time and human flow

a: Temperature

The green curve in Figure 11 shows the fluctuation in outdoor temperature. Except for the aboveground space of tp1, the temperature at other points was not directly affected by solar radiation; therefore, the temperature curve hardly varied over time. The overall temperature environment at the middle of the tubular space was the best, followed by the tube entrance space near the shopping center. The third best was the tube entrance space near the subway measurement point. The thermal environment of the station hall was the worst, maintaining the highest temperature nearly the entire day. Changes in the flow rate had little influence on the thermal environment of the tubular space; the relative position was the decisive factor for the temperature there.

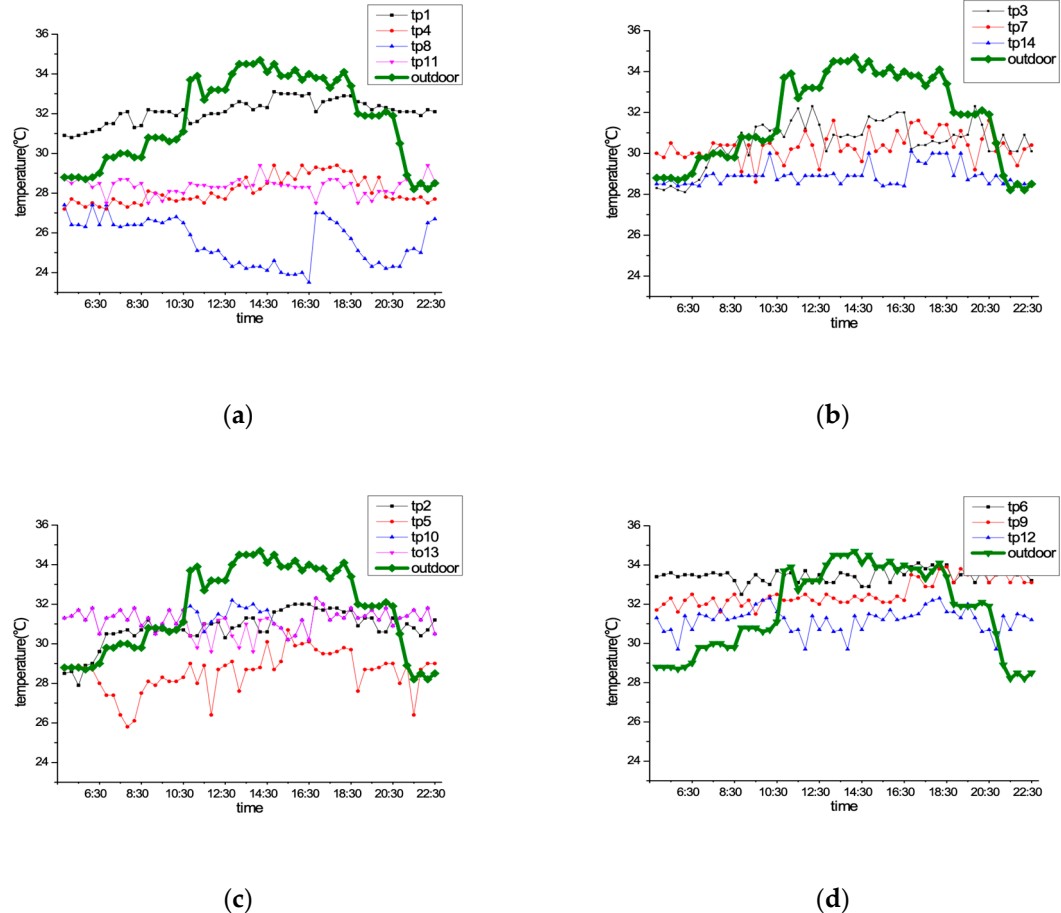

**Figure 11.** Comparison curves of hourly temperature data for each point during the monitoring day. (**a**) Middle tubular space; (**b**) Tube entrance (connection to building); (**c**) Tube entrance (connection to subway); (**d**) Station hall.

b: $CO_2$ concentration

In the diurnal variation curve for $CO_2$ concentration, there was almost no consistent fluctuation pattern for the same type of tubular space. From the overall curvilinear relation in Figure 12, it can be seen that there were two main trends. The first was the stable value for the whole day; the range of change was not large, as illustrated by test points tp8, tp13, and tp14. The second was the dramatic changes during two time periods, 8:00~9:00 a.m. and 4:30~7:30 p.m., which indicated that the concentration of $CO_2$ was significantly affected by indoor people flow, and the elevated concentration occurred during morning and evening peak hours. In addition, tp3, tp4, tp5, tp6, tp7, tp9, tp10, and tp12 exposed a lack of air adjustment capacity when a large number of people gathered together.

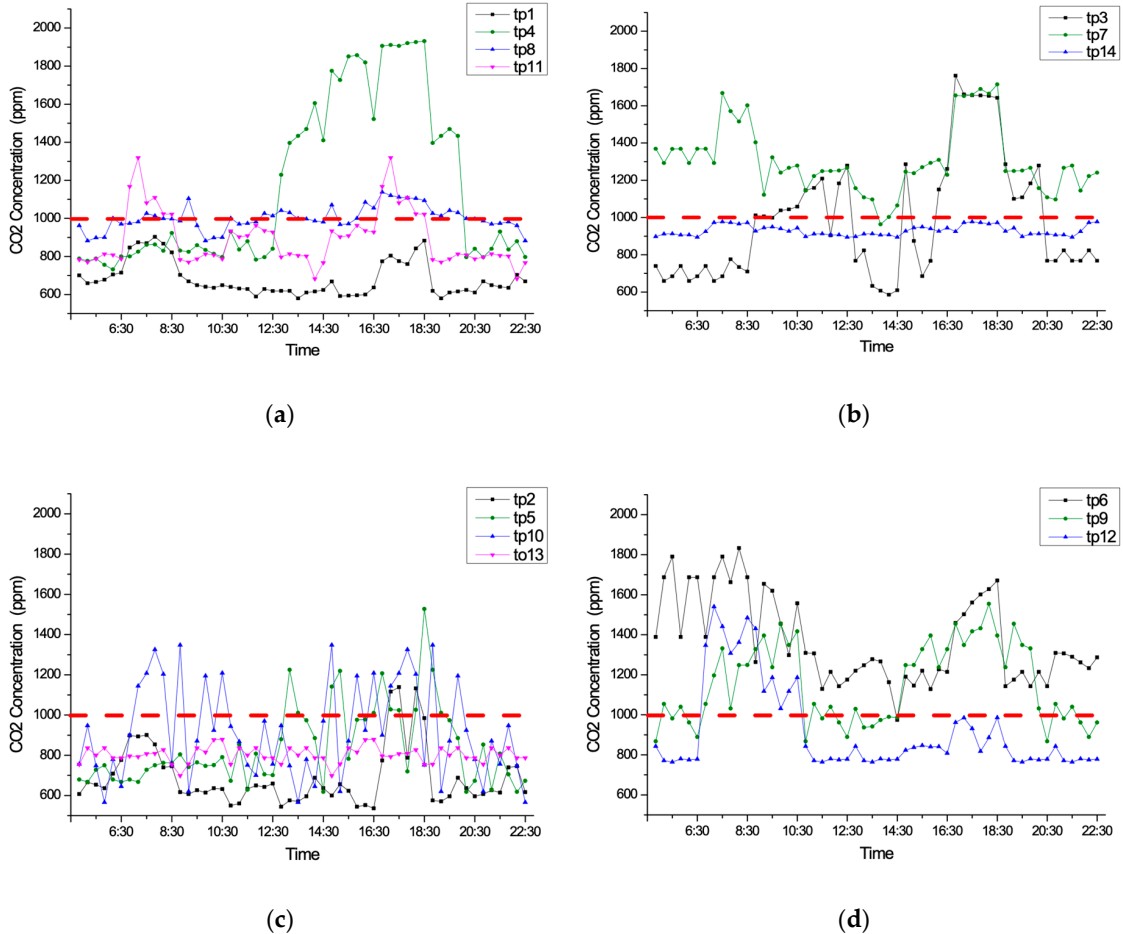

**Figure 12.** Comparison curves of hourly $CO_2$ concentration data for each point during the monitoring day. (**a**) Middle tubular space; (**b**) Tube entrance (connection to building); (**c**) Tube entrance (connection to subway); (**d**) Station hall.

c: PM2.5/PM10

As shown in Figure 13, the PM2.5 and PM10 data for each test point varied with time. This study took the middle of the tubular space to be the object of analysis. There were change rules over time for four test points: tp1, tp4, tp8, and tp11. The survey found that three points—tp1, tp4, and tp8—had a sudden increase in PM concentration at 11:00 a.m. and 6:00 p.m., almost two times that of other times. Tp11 was located in the tubular space near the #14 subway line, and was affected by the piston wind from the subway platform. The variations in PM concentration throughout the day were large, and the regularity presented was closely related to the subway's operation time.

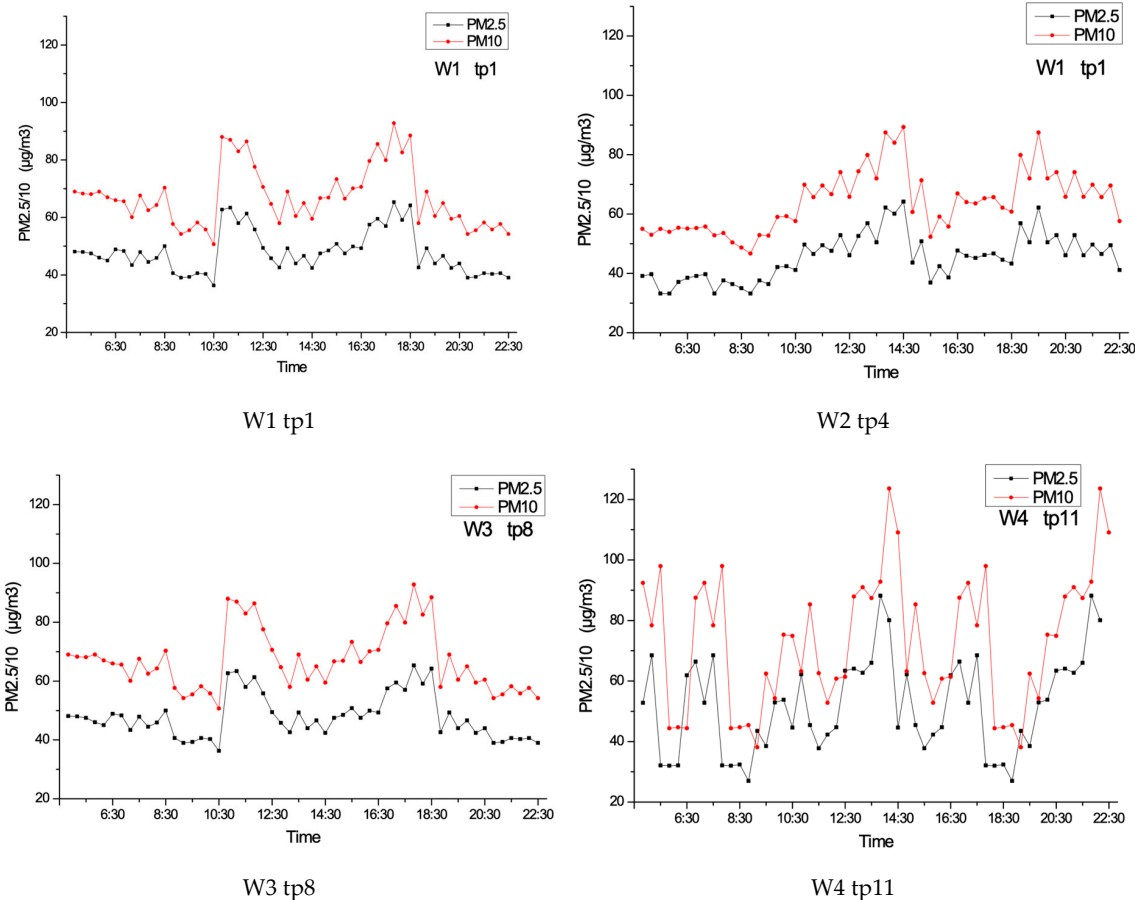

**Figure 13.** Comparison curves of hourly particulate matter 2.5 (PM2.5)/PM10 concentration data for each point during the monitoring day.

d: Wind speed

The test point height was 1.5 m (tp1, as shown in Figure 14a). One operating cycle for a train (arrival, stay, and departure) was around three minutes. Due to the piston effect caused by the train's operation, the tunnel wind reached remarkable levels. The wind speed when the train was arriving lasted for 15 s, with an average wind speed of 1.2 m/s. The stage during which the train stayed at the station lasted for 40~50 s (at the test point it was 48 s), with an average wind speed of 0.84 m/s. The train's departure stage lasted longer, about 120 s, with a maximum instantaneous wind speed of up to 3.6 m/s and an average wind speed of 1.79 m/s. During each three-minute cycle, the maximum wind speed was 3.6 m/s during the departure stage, while the minimum wind speed was 0 m/s during the period when the train stayed at the station (as shown Figure 14b). The wind direction was opposite during the arrival and departure stages. These two wind directions were off set when the train stayed at the station, presenting a brief state of calm. According to the coupling experiment in the Beijing subway conducted by Mingliang Ren and others, the maximum wind speed near the train could reach 7.5 m/s at the test point height of 2 m [56] (tp2, as shown in Figure 14a).

3.2.2. Occupancy satisfaction survey results and analysis

The subjective occupancy survey was comprised of two parts: on-site questionnaires and on-site interviews. The subjective questionnaire adopted a 7-point scale, where −3 corresponded with "Very Dissatisfied", 3 referred to the respondent being "Very Satisfied", and 0 was neutral (as shown in Table 10). The subjective questionnaire was issued at each test point (tp1 to tp14), 40 copies each, for a total of 560 copies. There was a total of 551 valid questionnaires completed, for a recovery rate of 98%. The test period was selected to be the same as the physical environment test, the highest temperature

period for the Beijing area, from 28 June 2017, to 20 July 2017. The purpose of this research was to investigate the performance of the tubular space in terms of occupancy satisfaction under the most unfavorable conditions in the summer climate. In addition, the survey also included on-site interviews. Researchers talked with subway staff, retail traders, security guards, cleaners, and passers-by for a substantial period of time.

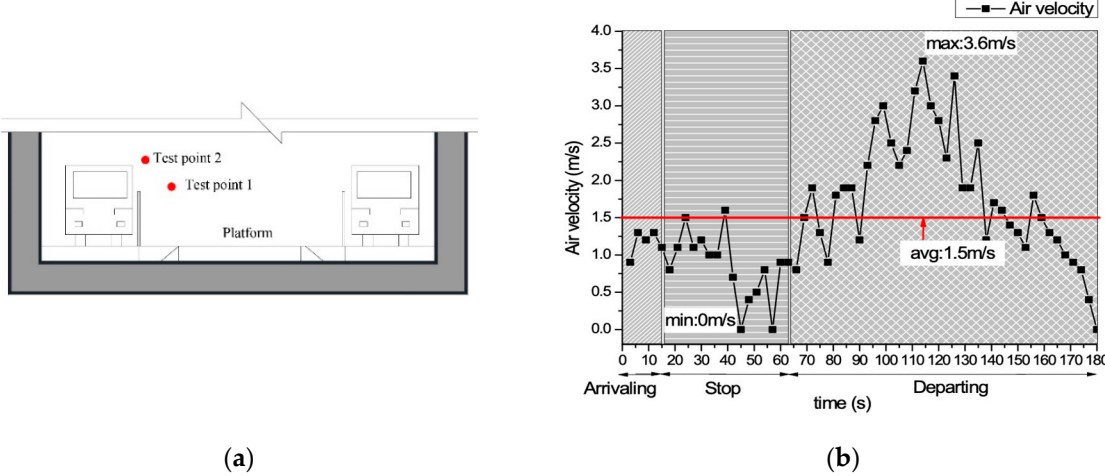

(**a**)  (**b**)

**Figure 14.** Wind speed curve for one operating cycle of a train. (**a**) Test point location; (**b**) Curve line.

**Table 10.** Questionnaire for the occupancy satisfaction survey.

| Occupancy Satisfaction | Very Dissatisfied | | Neutral | | Very Satisfied | | |
|---|---|---|---|---|---|---|---|
| Thermal comfort | | | | | | | |
| Humidity | | | | | | | |
| Air quality | | | | | | | |
| Lighting | | | | | | | |
| Ventilation | −3 ☐ | −2 ☐ | −1 ☐ | 0 ☐ | 1 ☐ | 2 ☐ | 3 ☐ |
| Ease of use | | | | | | | |
| Cleanliness and maintenance | | | | | | | |
| Overall environmental quality satisfaction | | | | | | | |

Table 11 shows the average data collected from the questionnaire. Corresponding with the analysis of the physical environment test results, this research also classified the 14 test points according to their location. There were a total of four types: middle tubular space, tube entrance (connection to building), tube entrance (connection to subway), and station hall.

Based on a histogram analysis of Figure 15, the researchers found that occupants had higher levels of satisfaction in the middle tubular space, and most of the collected data were positive. The problems with higher temperatures and poor lighting were clear in the physical environment test, but the subjective feelings of the users were not obvious. However, in terms of humidity, both the subjective questionnaire and the interviews reflected the occupants' discomfort.

The satisfaction results at the tube entrance (connection to building) fell into two categories. The first type was in relation to where the tube entrance connected the building to the outdoor space (tp3 and tp7). All indexes of satisfaction were low, especially in terms of thermal comfort, humidity,

air quality, and convenience. The other type related to where the tube entrance connected to the underground building (tp14), which demonstrated positive advantages for all indicators.

**Table 11.** Results data from occupancy satisfaction survey.

| Occupancy Satisfaction Vote | Tp1 | Tp2 | Tp3 | Tp4 | Tp5 | Tp6 | Tp7 | Tp8 | Tp9 | Tp10 | Tp11 | Tp12 | Tp13 | Tp14 |
|---|---|---|---|---|---|---|---|---|---|---|---|---|---|---|
| Thermal Comfort | −0.23 | 1.1 | −0.9 | 1.3 | −1.1 | −1.8 | −2.3 | 2.1 | −2.1 | −1.2 | 0.9 | −2.2 | −2.5 | 1.9 |
| Humidity | −0.25 | 0.2 | −1.5 | −1.6 | −2.5 | −2.2 | −2.2 | −0.6 | −2.4 | −1.5 | 0.8 | −2.3 | −2.1 | 1.5 |
| Air Quality | −0.15 | −0.3 | −1.2 | 0.2 | −1.9 | −2.6 | −1.9 | 1.1 | −1.5 | −1.2 | 1.1 | −2.3 | −0.5 | 1.3 |
| Lighting | 2 | −0.9 | 0.5 | 0.3 | −0.8 | 0.2 | −0.6 | 1.2 | 0.8 | 0.8 | 1.2 | −0.5 | 0.2 | 1.5 |
| Wind Comfort | −0.9 | 1.1 | −0.6 | 0.5 | −2.6 | −0.5 | −0.5 | 1.4 | 0.5 | −1.6 | 1.2 | −1.1 | −2.4 | −0.1 |
| Ease of Use | −0.15 | 0.6 | −2.1 | 1.1 | 0.2 | 0.9 | 1.1 | 0.9 | 0.6 | −0.2 | 1.1 | 0.3 | 0.2 | 0.2 |
| Cleaning and Maintenance | 0.6 | 0.5 | 1.6 | 0.5 | 1.8 | 1.2 | −1.6 | 1.3 | 0.7 | 0.2 | 1.3 | 0.2 | 0.3 | 0.3 |
| Overall Environmental Quality | 0.5 | 0.6 | 1.1 | 0.5 | −0.9 | 0.4 | −1.2 | 1.4 | 0.5 | 0.3 | 1.3 | −1.2 | −0.9 | 0.8 |
| Overall | 0.18 | 0.36 | −0.39 | 0.35 | −0.98 | −0.55 | −1.15 | 1.10 | −0.36 | −0.55 | 1.11 | −1.14 | −0.96 | 0.93 |

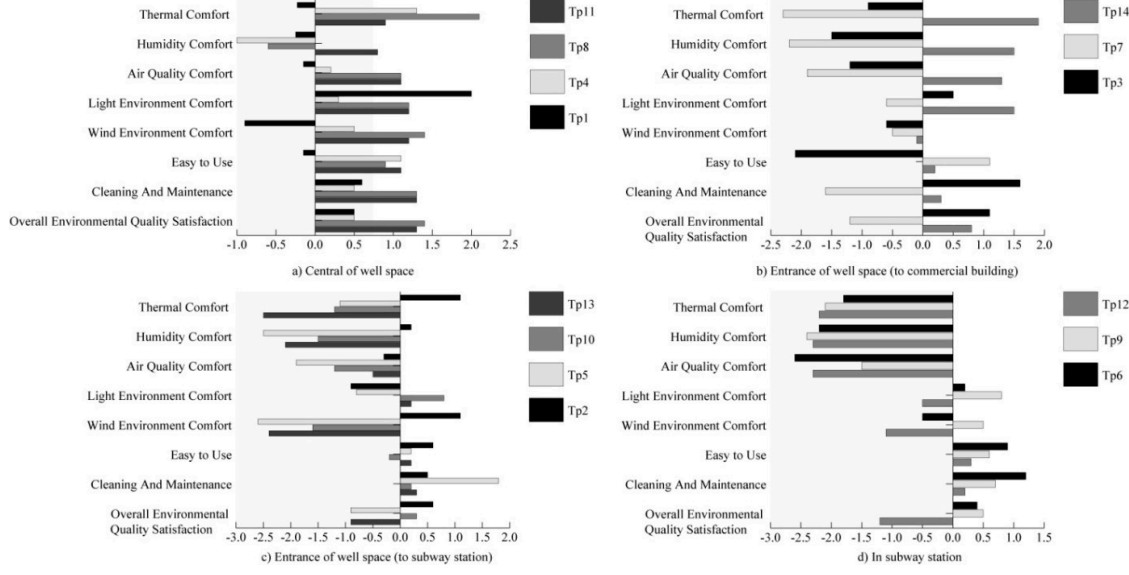

**Figure 15.** Results analysis of the occupancy satisfaction survey.

The most complaints from occupants were from the area near the subway. Both the space entrance near the subway and the subway platform level received lower occupancy satisfaction scores. The average thermal comfort of the tube entrance space near the subway was −1.85, the humidity score was −1.48, and air velocity comfort score was −1.38. Compared with the physical environment test results, the thermal environment of the tube entrance space near the subway had the greatest influence on the physiology and psychology of the human body. It was also the weakest point of the inner tubular space. In the physical environment test, the performance of the station hall was the worst, which is reflected in its high temperature, humid environment, and poor air quality. The averages of the three categories from the subjective survey data—thermal environment, humidity, and air quality—were as follows. Tp6 was −2.2, Tp9 was −2, and tp12 was −2.3, which indicates extreme occupant dissatisfaction.

All of the questionnaire items can be sorted as follows: tp11 (middle tubular space) > tp8 (middle tubular space) > tp14 (tube entrance type1) > tp2 (tube entrance type2) > tp4 (middle tubular space) > tp1 (middle tubular space) > tp9 (station hall) > tp3 (tube entrance type1) > tp6 (station hall) = tp10

(tube entrance type2) > tp13 (tube entrance type2) > tp5 (tube entrance type2) > tp12 (station hall) > tp7 (tube entrance type 1).

### 3.2.3. Satisfaction–Comfort Matrix Results and Analysis

Figure 16 shows the results of the Satisfaction–Comfort Matrix in the overall 14 target test points. The horizontal axis of the matrix corresponds to the level of each comfort (thermal, lighting, ventilation, and air quality) with the physical environment inside of the tubular space. The data was calculated into a comfort percentage obtained from the test results, and divided into six classes from 0 to 100%. The vertical axis of the matrix corresponds to the subjective analysis of occupancy space satisfaction, following with the −3~3 score scale outlined in the above research method.

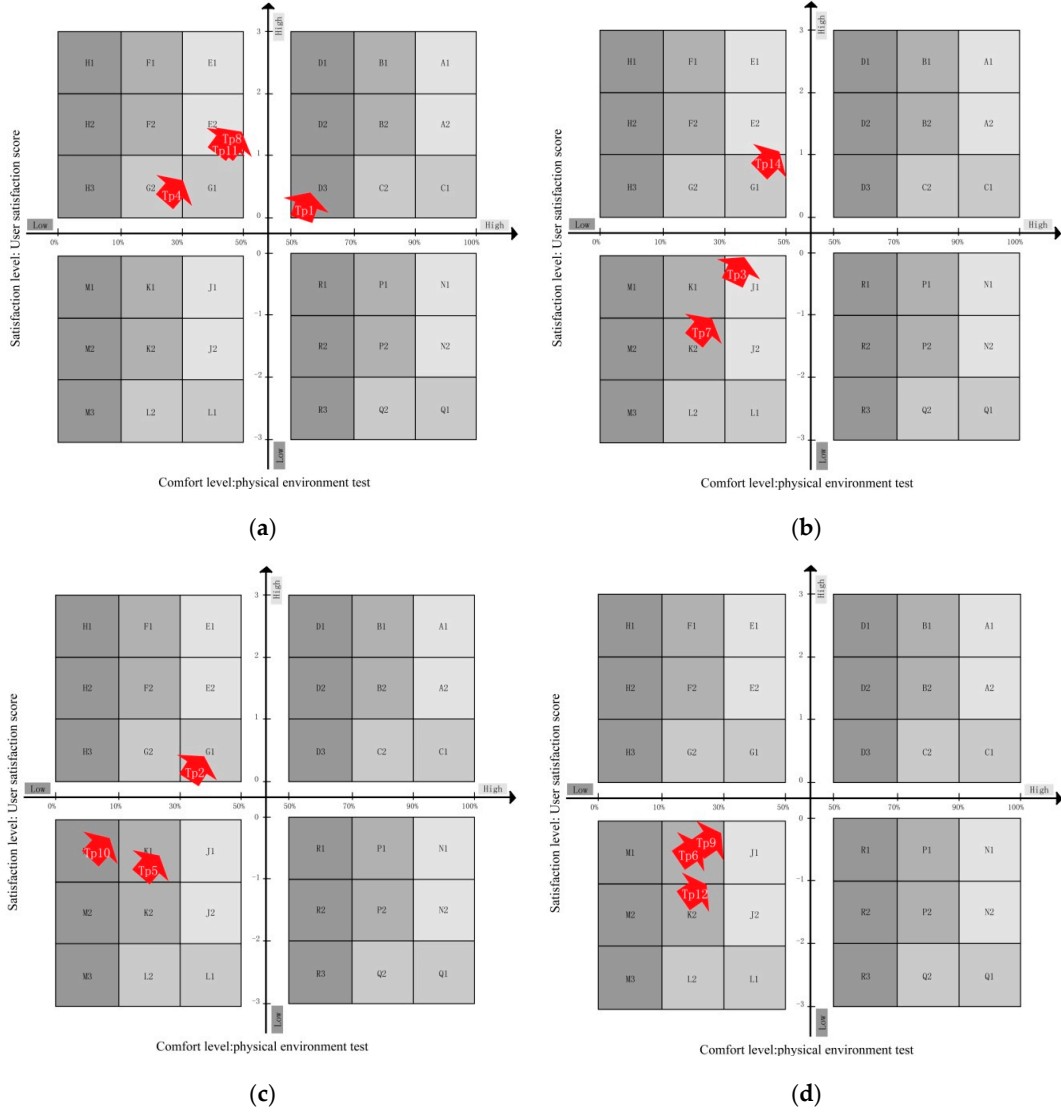

**Figure 16.** Test Point Satisfaction-Comfort Matrix results. (**a**) Middle tubular space; (**b**) Tube entrance (connection to building); (**c**) Tube entrance (connection to subway); (**d**) In subway station.

The Satisfaction–Comfort Matrix is divided into four quadrants. The results located in the first quadrant indicate that the animate space has a positive effect in terms of both satisfaction and comfort. The results located in the second quadrant mean it has a positive effect in terms of satisfaction but a negative effect in terms of comfort. The results located in the third quadrant imply neither satisfaction nor comfort. The results located in the fourth quadrant mean it has a negative effect in terms of

satisfaction but a possible positive effect in terms of comfort [39]. Each quadrant's evaluation is divided into four grades (as shown in Figure 16).

### 3.3. Problem Analysis and Potential

According to the investigation of the objective physical environment and subjective comfort feelings, it can be concluded that the potential for passive utilization of tubular space includes two aspects: Improvement in comfort and utilization of climate.

### 3.3.1. Comfort Improvement from the Perspective of Passive Space Design

Tube entrance space plays a role in cohesion and intermediary conversion. However, it was revealed that in the Beijing subway station building complex in the underground tube entrance convergence space, the instantaneous wind speed was too high in winter and transition seasons; this seriously affected user comfort and the long-term health of security personnel. Combining a design for the complex building's tube entrance transitional buffer space with setting a reasonable wind barrier along the narrow tube entrance area, as tube as identifying transfer space in the underground connected tubular space, would help to avoid the local wind outlet issue (as shown in Figure 17).

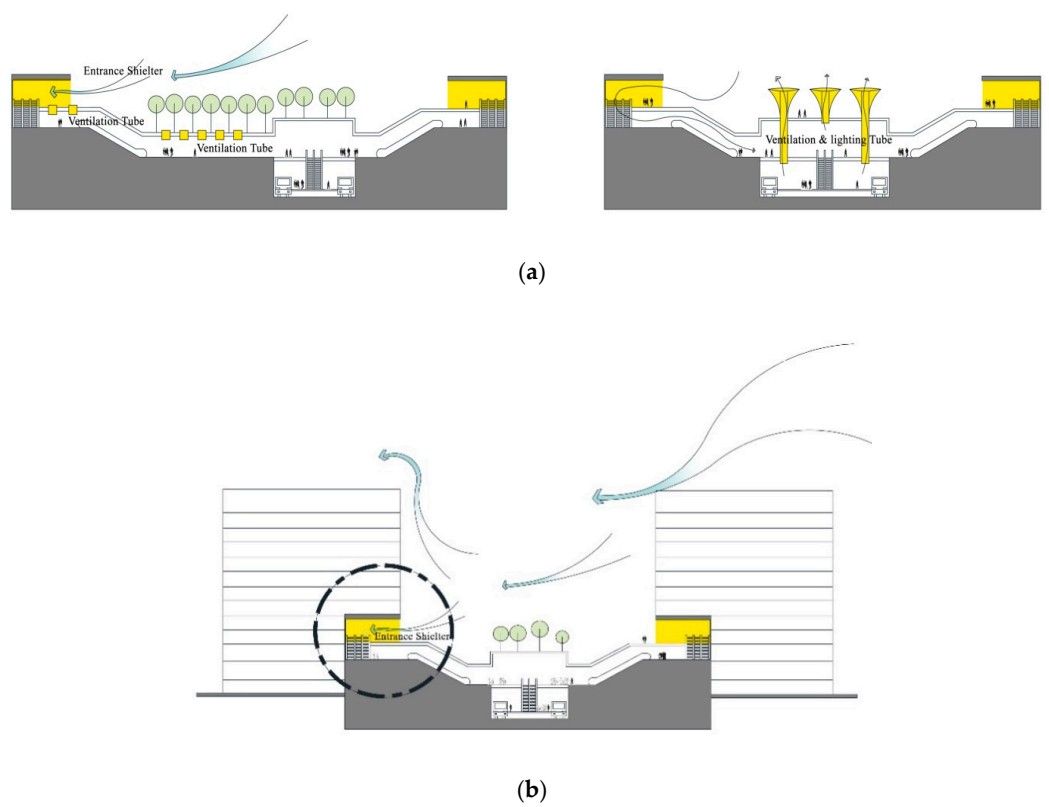

(**a**)

(**b**)

**Figure 17.** Passive strategy for relieving excessive wind velocity at the tube entrance. (**a**) Setting conversion space in the underground interconnected tubular space; (**b**) Transition buffer space at the tube entrance that combines with the complex building design.

The tubular space in the subway station area is always long and narrow. People in these locations experience high traffic flow, and the space lacks connection with the outdoor natural environment; this often results in poor air quality. Because it is a pass-through space, it is regularly neglected. Considering the complex building shape, setting aside areas such as air shafts, light tubes, atriums, and sunken courtyards would be an effective way of improving the air quality and indoor environment through space-based strategies.

### 3.3.2. Climate Utilization from the Perspective of Passive Space Design

However, tubular space as a means of connecting the external and internal environments has the advantage of transmitting natural resources. Effective natural resources can be transferred to the ground and underground spaces in subway transportation areas to solve comfort-based shortcomings in tubular spaces. These spaces can improve the quality of the area and reduce possible energy consumption during operation. In subway station building complexes, using aerodynamic, piston, and mixed ventilations, and exporting hot air to the outside through ventilation shafts or entrances can reduce energy consumption from air conditioning equipment, as tube as the area's levels of heat and humidity. Research has explored new modes for composite space systems, such as tubes with wind tunnels and solar chimneys [57], air supply tunnels that displace wind for better ventilation, active and passive combinations of wind tunnels and ground source heat pumps, and lighting from hot-press ventilation channels. These are all typical spatial patterns that can improve the utilization efficiency of passive tubular space. In addition, the shape of the tubular space has the advantage of introducing natural light. Natural light can improve the comfort level of the light environment and reduce lighting energy consumption. According to the actual test data, in a summer outdoor temperature of 28 °C, the air temperature in a space covered by four meters of soil can be maintained at 10 °C. When the outdoor temperature drops to −5 °C, the air temperature in that same underground area will remain stable at 10 °C [58]. In subway station building complexes, using geothermal energy with tubular spaces will not only improve the thermal environment quality and passive space design, but also transfer comfortable temperatures to other functional areas surrounding the tubular space, improving comfort and reducing the cost of operation.

### 3.4. Design Target and Space Conception of Tubular Composite Spaces from the Perspective of Sustainable Development

#### 3.4.1. Passive Design for High Performance and Low Energy Consumption-Oriented Tubular Spaces in Subway Station Building Complexes

It was determined that there is often low levels of comfort and user satisfaction problems with tubular space. To design a new integrated subway station building complex both at home and abroad, one must continuously attempt to optimize the indoor environment and further coordinate the design strategy with the setting. Thus, this research studied tubular spaces in subway station building complex integration designs, forming a passive design method for tubular spaces that would offer high levels of performance and low amounts of energy consumption. This high level of performance is expressed in greater comfort and improved human health related to the indoor thermal environment, light and air quality, and user experience. Low energy consumption comes in the form of an energy-saving role in the building, such as by providing passive cooling, a fresh air supply, natural light, and wind energy utilization via the functional and shape advantages of tubular space (as shown in Figure 18).

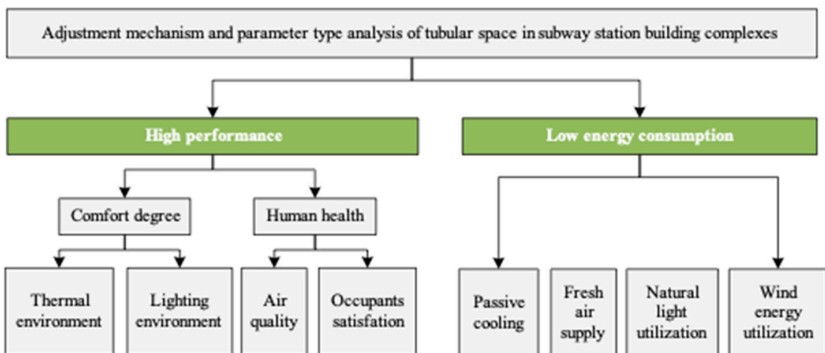

**Figure 18.** Type analysis of the passive parameters of high performance and low energy consumption-oriented tubular space in subway station building complexes.

### 3.4.2. Three Typical Design Concepts for Composite Tubular Space

An integrated design concept was presented in the study; it included two key aspects: the integration of an independent original design division that considered the organization of a single building, the underground station, and station space in the subway station building complex design process; and the integration of the space system with the goals of high performance and low energy consumption, to optimize the comprehensive performance of the composite space. Based on these issues, three design concepts for composite spaces are described below. The verification of their performance will be analyzed in detail in future research.

(a) Tubular complex space system for a wind tunnel and solar chimney (building atrium) for air filtration

The subway station building complex is a relatively complicated building space. Because of its large area, functional space coordination, and significant number of users, atriums are usually employed to organize the functional relationship. Vertically high atrium spaces can serve as a draft for solar chimneys. Combining them with a horizontal tunnel shaft design will help improve the quality of the indoor space's thermal environment and reduce the building's amount of energy consumption. In terms of the thermal environment, tunnel air temperature tends to be low, and combining the low temperature of tunnel air with air conditioning will reduce energy consumption in summer. Combining wind tunnels with the construction of solar chimneys and using the air dynamics principle of cold air sinking and hot air rising will benefit passive air circulation in buildings, exhausting high-temperature air and sucking low-temperature air from the tunnels. In addition, moderate thermal pressure ventilation in buildings is beneficial to the circulation of air and can reduce surface body temperature in summer, thus improving thermal comfort. However, due to the wind tunnels the air quality will remain low. Therefore, the composite tubular space system should be combined with an air filtration system to improve the air quality inside the tube and air circulation throughout the subway station building complex (as shown in Figure 19).

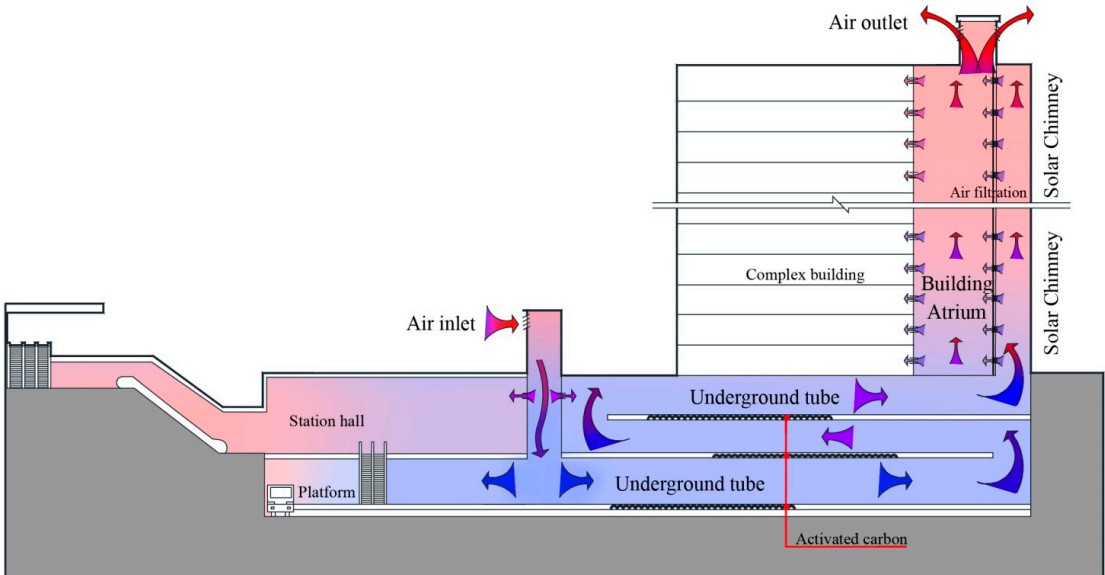

**Figure 19.** Sketch map of a complex tubular space system for wind tunnel and solar chimney air filtration.

(b) Complex tube path space system for lighting using hot air ventilation

The types of tubular space in a building can be divided into vertical, horizontal, and mixed. The advantage of a vertical tubular space is that it penetrates space in a vertical direction, which is beneficial for energy flowing from the top to the bottom, or vice versa. Natural lighting is a scarce resource in underground space, and thus underground areas consume much more energy for lighting

than do general ground-level buildings. In most subway station building complex buildings, atrium spaces or pass heights are the most common organizational structures. The integrated design idea connects building complexes and subway station halls to provide natural lighting in tubular space, forming light tubes with the atrium space. In addition, vertical tubes in atriums can offer useful hot air ventilation. The low air temperature of tunnel wind can result in a substantial temperature difference from the higher air temperatures at the tops of atriums, which can lead low-temperature air into the building. This composite space system can also integrate the composite tubular space system of a wind tunnel with a solar chimney (atrium) for air filtration, and integrate natural light with the air filtration system, ventilation, and other functions resulting in a composite function space. This yields further improvement in the passive space adjustment effect (as shown in Figure 20).

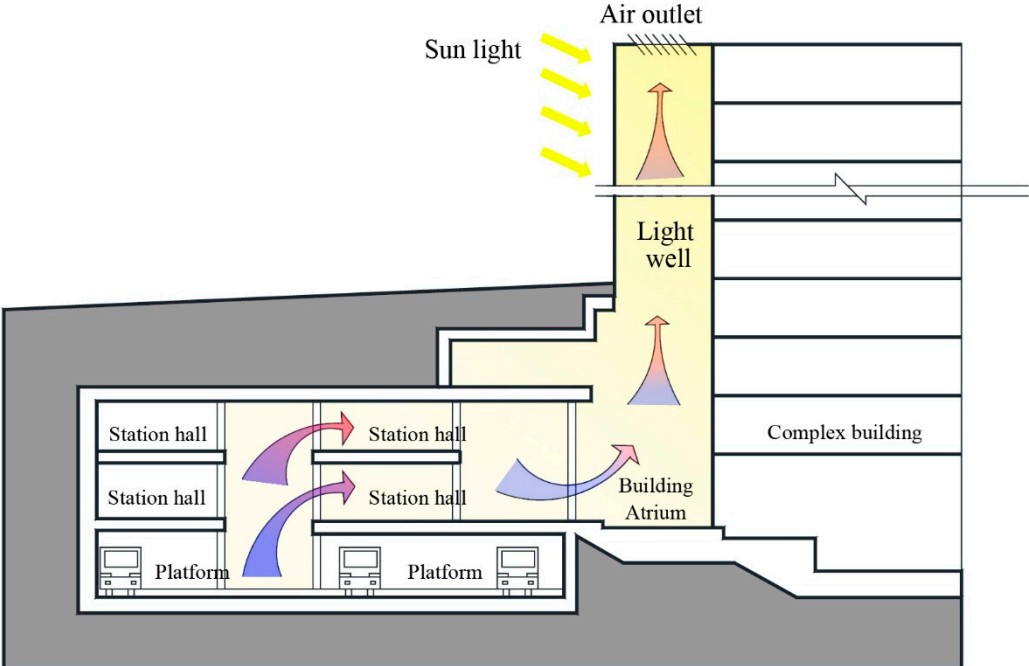

**Figure 20.** Sketch map of lighting through hot pressing the ventilation in a composite space system.

(c) Piston wind power generation for an auxiliary lighting system

According to past research, if a large number of fans are installed in subway tunnels, the blades produce air resistance, resulting in resistance to the train operation. This can lead to more energy consumption. However, if a few small micro-sized wind power generators are laid out in the subway station, the machines will generate renewable energy for the platform, station hall layer, and artificial lighting for the complex connection space. This is a useful way of using wind energy in subway tunnels. According to the survey distributed for this research, the average wind speed at a height of 1.5 m above the platform is 1.5 m/s, and the average wind speed near the upper end of the metro vehicle can reach 7.5 m/s. Previous research has shown that large, lightweight twist blades are the best choice for subway tunnels. In micro-sized wind turbines, the starting wind speed for vertical axis force generator equipment is 1 m/s, and the rated wind speed is 11 m/s. The safe velocity is 45~60 m/s, and the rated power is 200 W, 300 W, or 400 W. If the subway's daily schedule runs from 6:00 a.m. to 10:00 p.m., the system will be in operation for 16 hours a day. If the 400 W fan is selected, two units can be installed at the corner of each platform floor (for a total of eight units). This would result in 51.2 KW·h of electricity sent per day for artificial lighting demands. According to the general industrial and commercial electricity fee collection standards in Beijing, the annual payback period for each of these typhoon machines would be two to three years, with an average value of 1 Rmb/degree (as shown in Table 12).

**Table 12.** Economic benefit calculation for piston wind power generation for an auxiliary lighting system.

| Rated Power | Impeller Diameter | Lump Sum Investment | Additional Investment | Service Life | Annual Maintenance Cost | Daily Power Output | Annual Power Output | Payback Period of Investment |
|---|---|---|---|---|---|---|---|---|
| 200 | 0.47 | 2000 | 1200 | 10 | 100 | 3.2 | 1100 | 2.99 |
| 300 | 0.66 | 2100 | 1500 | 10 | 150 | 4.8 | 1750 | 2.24 |
| 400 | 0.66 | 2400 | 2000 | 10 | 300 | 6.4 | 2300 | 2.16 |

| Note | | |
|---|---|---|
| | 1. | A one-time investment would be the total cost of the system's installation. |
| | 2. | Additional investments would be used to replace two battery charges for 10 years of life. |

## 4. Conclusions

This study focused on the various tubular space forms in subway station building complexes; the goal was to identify those that would have a moderating effect on passive potential space. The result of this investigation, research, and analysis was an improvement of the indoor environment in terms of comfort and energy consumption. This work followed four key pathways.

(1) It attempted to establish a scientific and logical method for verifying the value of tubular space by establishing causal relationships among the parameterized building space information factors, occupancy satisfaction elements, physical environment comfort aspects, and climate conditions. This research adopted an analytical hierarchical process methodology for classifying each quantized building information factor, and then compared this information to the actual fieldwork physical environment test and occupant satisfaction vote data, in order to discern the key advantages and weaknesses in tubular space design in subway station building complexes.

(2) Based on the actual field investigation, a database of physical environment performance data and users' subjective satisfaction information was established. The results showed that 59% and 57.4% time in in middle tubular space was out of thermal and humidity comfort zone, and the uncomfortable temperature and humidity at the tube entrance measuring point reached 100%. Air quality was the worst in middle tube space, 65% of the time is unhealthy which exposing significant environmental problems. By analyzing correlations therein, the comfort and health problems found in different locations that were related to tubular spaces, as tube as the potential for passive utilization, were able to be discussed.

(3) According to the investigation of the objective physical environment and subjective comfort feelings, it can be concluded that the potential for passive utilization of tubular space includes two aspects: improvement in comfort level itself and utilization of climate between natural or artificial. Based on the measured data, an integrated design method for tube path spaces exhibiting high levels of performance and low amounts of energy consumption in subway station building complexes was put forward.

(4) The research described three typical composite tubular space designs, including wind tunnels/solar chimneys (atriums), which are composite tubular spaces with air filtration systems; composite space systems that provide lighting by hot-pressing the ventilation; and piston wind power generation for use in auxiliary lighting systems. These provide methods and ideas for future research and design.

**Author Contributions:** Conceptualization, J.L.; methodology, J.L.; software, S.L.; validation, Q.W.; formal analysis, J.L.; investigation, J.L., S.T., Y.J.; data curation, J.L.; writing—original draft preparation, J.L.; writing—review and editing, S.L.; visualization, Q.W.; project administration, J.L.; funding acquisition, J.L.

**Funding:** This work was supported by the Fundamental Funds for Beijing Natural Science Foundation of China (Grant No. 8182043), and the National Natural Science Foundation of China (Grant No. 51708019).

**Acknowledgments:** Bejing Jiaotong University; Beijing MTR Construction Administration Corporation.

**Conflicts of Interest:** The authors declare no conflict of interest.

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
