# Peer review of "Study of Passive Adjustment Performance of Tubular Space in Subway Station Building Complexes"

_applsci, doi:10.3390/app9050834_

Round 1
Reviewer 1 Report
The article is interdisciplinary and investigates many indicators. The authors did a good job in creating a readable and informative article despite this challenge.
The article can be accepted in its present form. Some suggestions for improvements are included below, however:
This reviewer has general knowledge about energy use in buildings, but is not well-versed in architectural terms. Nevertheless, I will point out that, in the English language, the term "well" normally refers to a shaft, i.e. a vertical corridor. The term "well-type space" in figure 1, however, includes both horizontal and vertical corridors. In addition, the referenced literature is apparently in Chinese or not easily available. This may therefore become a little confusing to the reader. Can the authors' understanding of "well-type space" in this context somehow be defined more clearly so that everyone understands what it is this early in the article, this will improve readability.
The formatting of tables 4, 5 and 11 could be improved.
Line 561: "is" always...
679: improvement "of"
Author Response
The article is interdisciplinary and investigates many indicators. The authors did a good job in creating a readable and informative article despite this challenge.
The article can be accepted in its present form. Some suggestions for improvements are included below, however:
This reviewer has general knowledge about energy use in buildings, but is not well-versed in architectural terms. Nevertheless, I will point out that, in the English language, the term "well" normally refers to a shaft, i.e. a vertical corridor. The term "well-type space" in figure 1, however, includes both horizontal and vertical corridors. In addition, the referenced literature is apparently in Chinese or not easily available. This may therefore become a little confusing to the reader. Can the authors' understanding of "well-type space" in this context somehow be defined more clearly so that everyone understands what it is this early in the article, this will improve readability.
Re: Thank you for your remind. In order to better conform to architectural terms, the full paper replaces "well-type space" with "tubular space". And the necessary description has been added from line 82-99.
line 82-99
“Tubular space includes horizontal and vertical corridors in buildings, usually in slender shape, such as ventilation shafts, patios and lighting wells, tunnel corridors for connection and so on. Tubular space occupies an important proportion of buildings in subway station building complexes, and its passive regulation has not been deeply investigated [23]. Subway station building complexes are affected by the characteristics of the mode of space utilization, wherein it is very common to use tubular space in ground-level and underground spaces (as shown in Figure 1), including patios and lighting wells to improve natural lighting efficiency. Other uses include ventilation shafts for improving the indoor thermal environment and air quality, a station's traffic tubes for connecting ground-level and underground stations, and tunnels for traffic transmission. Tubular space can be seen as a "communications device" that transmits people, mass, and energy to different spaces [24]. This space type is a passive adjustment strategy located between the external and interior environments of the building. It uses natural energy sources (such as wind, solar energy, and rainwater) and the natural environment to regulate microclimates and improve the indoor atmosphere. In subway station building complexes, tubular space has the potential to play an important role in passive adjustment performance, especially with regards to natural lighting and ventilation, passive cooling, etc., to optimize comfort and user satisfaction with the indoor space, and greatly reduce the energy consumption of the building’s operating phase [25].”
The formatting of tables 4, 5 and 11 could be improved.
Line 561: "is" always...
679: improvement "of"
RE:Thank you for your remind. The error has been corrected as shown in table 4,5, 11, line 561 and 679.

Reviewer 2 Report
Well-researched study presented clearly with excellent graphics.
Thorough description of research and analytical tools and methods strengthens observations and conclusions.
Author Response
Well-researched study presented clearly with excellent graphics.
Thorough description of research and analytical tools and methods strengthens observations and conclusions.
Re: Thank you for your remind. The expression and writing of English have been refined in full text.

Reviewer 3 Report
Dear Editor and authors,
the manuscript titled "Study of Passive Adjustment Performance of Well-2 type Space in Subway Station Building Complexes’’ tries to develop and apply a conceptual methodology for evaluating the passive performance in subway building complexes.
This study presents the methodology’s steps and the results from the field survey and tries to propose solutions-design concepts to improve the passive performance in these kind of well-type buildings. However, this study lacks in cohesion between the complex methodology with the results and the conclusions. The abstract is not helpful to understand what follows in the study. It is not very clear how the field survey helped in the proposed design concepts. The conclusions seem to be very general. It is a not easily reading manuscript while the terminology throughout the text is not very clear. The manuscript should be improved by updating the research and methodology design (complexity doesn’t always work) as well as the whole presentation of the study.
Author Response
Dear Editor and Reviewers:
Thank you for your comments concerning our manuscript entitled “Study of Passive Adjustment Performance of Well-type Space in Subway Station Building Complexes” (ID: applsci-434436). They are very helpful for paper revision. After a careful understanding and study, our responds to the comments are as follows.
All the changes to the text have been revised by using the "Track Changes" function in
Microsoft Word, and responses to all comments been highlighted in red.
Dear Reviewer 3
Open Review
(x) I would not like to sign my review report
( ) I would like to sign my review report
English language and style
(x) Extensive editing of English language and style required
( ) Moderate English changes required
( ) English language and style are fine/minor spell check required
( ) I don't feel qualified to judge about the English language and style
Yes | Can be improved | Must be improved | Not applicable | |
Does the introduction provide sufficient background and include all relevant references? | ( ) | (x) | ( ) | ( ) |
Is the research design appropriate? | ( ) | ( ) | (x) | ( ) |
Are the methods adequately described? | ( ) | ( ) | (x) | ( ) |
Are the results clearly presented? | ( ) | ( ) | (x) | ( ) |
Are the conclusions supported by the results? | ( ) | ( ) | (x) | ( ) |
Comments and Suggestions for Authors
Dear Editor and authors,
the manuscript titled "Study of Passive Adjustment Performance of Well-2 type Space in Subway Station Building Complexes’’ tries to develop and apply a conceptual methodology for evaluating the passive performance in subway building complexes.
1. This study presents the methodology’s steps and the results from the field survey and tries to propose solutions-design concepts to improve the passive performance in these kind of well-type buildings. However, this study lacks in cohesion between the complex methodology with the results and the conclusions.
Re: Thank you for your remind.
The structure of this research is as follows:
1. Introduction
1.1. Research background
1.2. Passive design of tubular space in subway station building complexes
1.3. Objective of this study
2. Methodology
2.1. Stage one: Factor analysis of the effect of passive design on tubular space
2.2. Stage two: Field survey
2.3. Stage three: Problem and analysis of the spatial potential
2.4. Stage four: Set up new target orientation and space update
3. Results and Discussion
3.1. Building space information factors
3.2. Field survey results and analysis
3.3. Problem analysis and potential
3.4. Design target and space conception of well pattern composite spaces from the perspective of sustainable development
4. Conclusions
The four steps of the research method in Section 2 correspond to the data analysis in Section 3 Results and Discussion. The specific research sequence is:
First, the factors that affect the passive adjustment performance were analyzed. Then, according to the analytic factors and taking the urban Beijing subway station building complexes as an example, a long-term physical environment test was carried out. The subject was an subway station building complex with a typical amount of tubular space. This research focused on the physical environment, as tube as the passive function of potential space and its influence on surrounding functional areas. It included actual measurement results such as the thermal conditions, air ventilation, lighting environments, indoor air quality, occupant satisfaction and comfort, and other subjective feedback. Through this objective investigation of the physical environment and subjective feedback of the occupants' degree of comfort, the problems with objective space were able to be studied and analyzed, and the potential for spatial optimization put forth from the perspective of passive adjustment performance. Finally, the database established through this research assisted in highlighting the design goals for tubular space in subway station building complexes. A model for three typical kinds of composite tubular spaces was constructed with the goal of achieving high levels of performance and low amounts of energy consumption.
2. Methodology | 3. Results and Discussion |
2.1. Stage one: Factor analysis of the effect of passive design on tubular space | 3.1. Building space information factors |
2.2. Stage two: Field survey | 3.2. Field survey results and analysis |
2.3. Stage three: Problem and analysis of the spatial potential | 3.3. Problem analysis and potential |
2.4. Stage four: Set up new target orientation and space update | 3.4. Design target and space conception of well pattern composite spaces from the perspective of sustainable development |
2. The abstract is not helpful to understand what follows in the study. It is not very clear how the field survey helped in the proposed design concepts.
Re: Thank you for your remind. The abstract was rewrite following the line of methodology.
The stereo integration of subway transportation with urban functions has promoted the transformation of urban space via extensive two-dimensional plans to intensive three-dimensional development. As sustainable development aspect, it has posed new challenges for the design of architectural space to be better environmental quality and low energy consumption. Therefore, subway station building complexes with high-performance designs should be a primary focus. Tubular space is a very common spatial form in subway station building complexes; it is an important space carrier for transmitting airflow and natural light. As such, it embodies the advantages of effectively utilizing natural resources, improving the indoor thermal and light environments, refining the air quality, and reducing energy consumption. This research took tubular space, which has a passive regulation function in subway station building complexes as its research object. It firstly established a scientific and logical method for verifying the value of tubular space by searching causal relationships among the parameterized building space information factors, occupancy satisfaction elements, physical environment comfort aspects, and climate conditions. Secondly, based on the actual field investigation, a database of physical environment performance data and users’ subjective satisfaction information was collected. Through the fieldwork results and analysis, the research thirdly concluded that the potential passive utilization of tubular space in subway station building complexes can be divided into two aspects: improvement in comfort level itself and utilization of climate between natural or artificial. Finally, three typical integrated design method for tubular spaces exhibiting high levels of performance and low amounts of energy consumption in subway station building complexes was put forward. This interdisciplinary research provides a design basis for subway station building complexes seeking to achieve high levels of performance and low amounts of energy consumption.
3 .The conclusions seem to be very general. It is a not easily reading manuscript while the terminology throughout the text is not very clear. The manuscript should be improved by updating the research and methodology design (complexity doesn’t always work) as well as the whole presentation of the study.
Re: Thank you for your remind. The conclusion has been revised, please refer to line 736-776.
Line 736-776:
This study focused on the various tubular space forms in subway station building complexes; the goal was to identify those that would have a moderating effect on passive potential space. The result of this investigation, research, and analysis was an improvement of the indoor environment in terms of comfort and energy consumption. This work followed four key pathways.
1) It attempted to establish a scientific and logical method for verifying the value of tubular space by establishing causal relationships among the parameterized building space information factors, occupancy satisfaction elements, physical environment comfort aspects, and climate conditions. This research adopted an analytical hierarchical process methodology for classifying each quantized building information factor, and then compared this information to the actual fieldwork physical environment test and occupant satisfaction vote data, in order to discern the key advantages and weaknesses in tubular space design in subway station building complexes.
2) Based on the actual field investigation, a database of physical environment performance data and users’ subjective satisfaction information was established. The results showed that 59% and 57.4% time in in middle tubular space was out of thermal and humidity comfort zone, and the uncomfortable temperature and humidity at the tube entrance measuring point reached 100%. Air quality was the worst in middle well space, 65% of the time is unhealthy which exposing significant environmental problems. By analyzing correlations therein, the comfort and health problems found in different locations that were related to tubular spaces, as well as the potential for passive utilization, were able to be discussed.
3). According to the investigation of the objective physical environment and subjective comfort feelings, it can be concluded that the potential for passive utilization of tubular space includes two aspects: improvement in comfort level itself and utilization of climate between natural or artificial. Based on the measured data, an integrated design method for well path spaces exhibiting high levels of performance and low amounts of energy consumption in subway station building complexes was put forward.
4) The research described three typical composite tubular space designs, including wind tunnels/solar chimneys (atriums), which are composite tubular spaces with air filtration systems; composite space systems that provide lighting by hot-pressing the ventilation; and piston wind power generation for use in auxiliary lighting systems. These provide methods and ideas for future research and design.

Reviewer 4 Report
The paper presents a study of passive adjustment performance of well-type space in subway station building complexes. The idea of the paper is interesting. However, before being published, I suggest some improvements. More information about the survey should be added (as example, information’s about the inquiries). Is preferred to add a table that include all data of the questionnaires. Include mean value, RMS and others. More information about the measuring system, included sensor information’s should be added. More information, as example schemes, about the space measured, included location of the sensors. In the study the thermal comfort conditions are used. However, in order to evaluate the thermal comfort conditions is important to evaluate the PMV and PPD indexes. These parameters are associated with air velocity, air temperature, air relative humidity, mean radiant temperature, activity level and clothing level. Using the different measured variables is interesting to evaluate the thermal comfort level. The variables were measured during the day. However only one instant survey was made. Is important to improve better this correlation. More information about the future developments is important to add. The conclusion should be improved in accordance with the obtained results.Author Response
Dear Editor and Reviewers:
Thank you for your comments concerning our manuscript entitled “Study of Passive Adjustment Performance of Well-type Space in Subway Station Building Complexes” (ID: applsci-434436). They are very helpful for paper revision. After a careful understanding and study, our responds to the comments are as follows.
All the changes to the text have been revised by using the "Track Changes" function in
Microsoft Word, and responses to all comments been highlighted in red.
Dear Reviewer 4
Open Review
(x) I would not like to sign my review report
( ) I would like to sign my review report
English language and style
( ) Extensive editing of English language and style required
( ) Moderate English changes required
(x) English language and style are fine/minor spell check required
( ) I don't feel qualified to judge about the English language and style
Re: Thank you for your remind. The expression and writing of English have been refined in full text.
Yes | Can be improved | Must be improved | Not applicable | |
Does the introduction provide sufficient background and include all relevant references? | ( ) | (x) | ( ) | ( ) |
Is the research design appropriate? | ( ) | ( ) | (x) | ( ) |
Are the methods adequately described? | ( ) | ( ) | (x) | ( ) |
Are the results clearly presented? | ( ) | ( ) | (x) | ( ) |
Are the conclusions supported by the results? | ( ) | ( ) | (x) | ( ) |
Comments and Suggestions for Authors
The paper presents a study of passive adjustment performance of well-type space in subway station building complexes. The idea of the paper is interesting. However, before being published, I suggest some improvements.
1. More information about the survey should be added (as example, information’s about the inquiries). Is preferred to add a table that include all data of the questionnaires. Include mean value, RMS and others.
Re: Thank you for your remind. All of physical environmental survey raw data has been upload as appendix (raw data 1-6). The test period was selected to be from June 28, 2017, to July 20, 2017, the highest-temperature time period in Beijing.
Occupancy satisfaction survey questionnaire was issued at each test point (tp1 to tp14), 40 copies each, for a total of 560 copies. There was a total of 551 valid questionnaires completed, for a recovery rate of 98%. The test period was selected to be the same as the physical environment test. the contents of questionnaire for the occupancy satisfaction survey please refer to table 10 and 11.
Table 10 Questionnaire for the Occupancy Satisfaction Survey.
Thermal comfort | Very Dissatisfied Neutral Very Satisfied -3 -2 -1 0 1 2 3 o o o o o o o |
Humidity | |
Air quality | |
Lighting | |
Ventilation | |
Ease of use | |
Cleanliness and maintenance | |
Overall environmental quality satisfaction |
Table 11 Results Data from Occupancy Satisfaction Survey .
Occupancy Satisfaction Vote | Tp1 | Tp2 | Tp3 | Tp4 | Tp5 | Tp6 | Tp7 | Tp8 | Tp9 | Tp10 | Tp11 | Tp12 | Tp13 | Tp14 |
Thermal Comfort | -0.23 | 1.1 | -0.9 | 1.3 | -1.1 | -1.8 | -2.3 | 2.1 | -2.1 | -1.2 | 0.9 | -2.2 | -2.5 | 1.9 |
Humidity | -0.25 | 0.2 | -1.5 | -1.6 | -2.5 | -2.2 | -2.2 | -0.6 | -2.4 | -1.5 | 0.8 | -2.3 | -2.1 | 1.5 |
Air Quality | -0.15 | -0.3 | -1.2 | 0.2 | -1.9 | -2.6 | -1.9 | 1.1 | -1.5 | -1.2 | 1.1 | -2.3 | -0.5 | 1.3 |
Lighting | 2 | -0.9 | 0.5 | 0.3 | -0.8 | 0.2 | -0.6 | 1.2 | 0.8 | 0.8 | 1.2 | -0.5 | 0.2 | 1.5 |
Wind Comfort | -0.9 | 1.1 | -0.6 | 0.5 | -2.6 | -0.5 | -0.5 | 1.4 | 0.5 | -1.6 | 1.2 | -1.1 | -2.4 | -0.1 |
Ease of Use | -0.15 | 0.6 | -2.1 | 1.1 | 0.2 | 0.9 | 1.1 | 0.9 | 0.6 | -0.2 | 1.1 | 0.3 | 0.2 | 0.2 |
Cleaning and Maintenance | 0.6 | 0.5 | 1.6 | 0.5 | 1.8 | 1.2 | -1.6 | 1.3 | 0.7 | 0.2 | 1.3 | 0.2 | 0.3 | 0.3 |
Overall Environmental Quality | 0.5 | 0.6 | 1.1 | 0.5 | -0.9 | 0.4 | -1.2 | 1.4 | 0.5 | 0.3 | 1.3 | -1.2 | -0.9 | 0.8 |
Overall | 0.18 | 0.36 | -0.39 | 0.35 | -0.98 | -0.55 | -1.15 | 1.10 | -0.36 | -0.55 | 1.11 | -1.14 | -0.96 | 0.93 |
2. More information about the measuring system, included sensor information’s should be added.
Re: Thank you for your remind. please refer sensor information in rightmost column in Table 2.
Table 2 Building the Physical Environment Fieldwork Test Framework.
Measurement Items | Parameter Type | Test Content | Properties of the instruments | |
Thermal environment | outdoor temperature test | temperature | ℃ | Portable infrared temperature meter, Biaozhi GM700, Range: -50~700℃, Resolution: 0.1℃ |
indoor temperature test for each (selected) test point | ||||
indoor humidity test for each (selected) test point | humidity | % | ||
Lighting | outdoor luminance test | luminance | lux / daylight factor % | Portable luminance meter, Reggiani DT-1301, Range: 0~50000 Lux, Resolution: 1 Lux |
indoor luminance test for each (selected) test point | ||||
IAQ | outdoor CO2 concentration test | CO2 concentration | ppm | Portable and self-record CO2 meter, TJHY-EZY-1, Range: 0~5000ppm, Resolution: 1ppm |
indoor CO2 concentration test for each (selected) test point | ||||
outdoor PM2.5/10 concentration test | PM2.5/10 concentration | μg/m³ | Portable air quality meter, temopt LKC-1000S+, Range: 0~999 mg/m3, Resolution: 0.01mg/m3 | |
indoor PM2.5/10 concentration test for each (selected) test point | ||||
indoor HCHO concentration test for each (selected) test point | HCHO | g/cm³ | ||
Ventilation | indoor air velocity test for each (selected) test point | air velocity | m/s | Self-record instrument for wind velocity and wind temperature, TJHY-FB-1, Range: 0~10M/S, Resolution: 0.01M/S |
indoor air temperature of each (selected) test point | air temperature | ℃ | Self-record instrument for environment, TJHY-HCZY-1, Range: 0~5000ppm, Resolution: 1ppm | |
3. More information, as example schemes, about the space measured, included location of the sensors.
Re: Thank you for your remind. the location of the sensors please refer more information from line 316-327 and figure 5.
Line 316-327:
“The researchers selected three or four test points for each site station; all tests contained 18 measuring points. Each test point had a certain representativeness. The W1 site contained three test points; tp1 was located in the middle of an above-ground glass corridor between the station and the complex, and was a middle tubular space.Tp4 and tp11 were located between the sites and underground associated wells, which were all classified as middle tubular space.Tp8 was a link between a subway station and commercial building, which was also middle tubular space.Tp3, tp7, and tp14 belonged to the first kind of tube entrance space, and connected the commercial complex building at one end of the well shaft.Tp2, tp5, tp10, and tp13 belonged to the second kind of tube entrance space, and connected to the station hall.Tp6, tp9, and tp12 were the test points at the subway station hall (as shown in Figure 5).Finally, tp15, tp16, tp17, and tp18 were the subway platform test points for W1, W3, W4, and W5, respectively.”
Fig.5 Test station space plan and test points location.
4. In the study the thermal comfort conditions are used. However, in order to evaluate the thermal comfort conditions is important to evaluate the PMV and PPD indexes. These parameters are associated with air velocity, air temperature, air relative humidity, mean radiant temperature, activity level and clothing level.
Re: Thank you for your remind. PMV and PPD indexes are associated with air velocity, air temperature, air relative humidity, mean radiant temperature, activity level and clothing level. The research had involved air velocity, air temperature, air relative humidity. Because the well space is located underground, there is no solar radiation, therefore, mean radiant temperature equals to air temperature.
Activity level and clothing level have been added in the text, please refer to line 342 to 345.
Line 342-345:
Tubular space in subway station building complexes is mostly underground, so the influence of radiation temperature can be ignored and people's metabolic rates can be set to the same level of 1.5 met[41]. During the test period, the hottest period in summer in Beijing was selected, so clothing level was chosen as 0.35 clo[41].
5. Using the different measured variables is interesting to evaluate the thermal comfort level. The variables were measured during the day. However only one instant survey was made. Is important to improve better this correlation.
Re: Thank you for your remind. The test period was selected to be from June 28, 2017, to July 20, 2017, the highest-temperature time period in Beijing. It is a typical and continuous testing period of 3 weeks. The data excluded unstable factors which may conduct to instantaneous mutations data such as weather mutations, active equipment interference, people behavioral interference, misuse of testing instruments, etc, and used a mean value within the 3 weeks.
please refer to revised text from line 305-311.
6. More information about the future developments is important to add. The conclusion should be improved in accordance with the obtained results.
Re: Thank you for your remind. The conclusion has been revised, please refer to line 727-766.
Line 727-766:
This study focused on the various tubular space forms in subway station building complexes; the goal was to identify those that would have a moderating effect on passive potential space. The result of this investigation, research, and analysis was an improvement of the indoor environment in terms of comfort and energy consumption. This work followed four key pathways.
1) It attempted to establish a scientific and logical method for verifying the value of tubular space by establishing causal relationships among the parameterized building space information factors, occupancy satisfaction elements, physical environment comfort aspects, and climate conditions. This research adopted an analytical hierarchical process methodology for classifying each quantized building information factor, and then compared this information to the actual fieldwork physical environment test and occupant satisfaction vote data, in order to discern the key advantages and weaknesses in tubular space design in subway station building complexes.
2) Based on the actual field investigation, a database of physical environment performance data and users’ subjective satisfaction information was established. The results showed that 59% and 57.4% time in in middle tubular space was out of thermal and humidity comfort zone, and the uncomfortable temperature and humidity at the tube entrance measuring point reached 100%. Air quality was the worst in middle well space, 65% of the time is unhealthy which exposing significant environmental problems. By analyzing correlations therein, the comfort and health problems found in different locations that were related to tubular spaces, as well as the potential for passive utilization, were able to be discussed.
3). According to the investigation of the objective physical environment and subjective comfort feelings, it can be concluded that the potential for passive utilization of tubular space includes two aspects: improvement in comfort level itself and utilization of climate between natural or artificial. Based on the measured data, an integrated design method for well path spaces exhibiting high levels of performance and low amounts of energy consumption in subway station building complexes was put forward.
4) The research described three typical composite tubular space designs, including wind tunnels/solar chimneys (atriums), which are composite tubular spaces with air filtration systems; composite space systems that provide lighting by hot-pressing the ventilation; and piston wind power generation for use in auxiliary lighting systems. These provide methods and ideas for future research and design.

Round 2
Reviewer 3 Report
Dear Authors,
thank you for your effort and for responding to all my comments.
Reviewer 4 Report
In the actual version, in general, all suggested by the reviewer were implemented.